# Regularizing Hidden States Enables Learning Generalizable Reward Model for LLMs

**Rui Yang**[1]  **Ruomeng Ding**[2]  **Yong Lin**[3 4]  **Huan Zhang**[1]  **Tong Zhang**[1]

[1]University of Illinois Urbana-Champaign, [2]Georgia Institute of Technology,
[3]Princeton University, [4]Princeton Language and Intelligence
`yangrui.thu2015@gmail.com, rmding@gatech.edu, yl7690@princeton.edu`
`huan@huan-zhang.com, tongzhang@tongzhang-ml.org`

## Abstract

Reward models trained on human preference data have been proven to effectively align Large Language Models (LLMs) with human intent within the framework of reinforcement learning from human feedback (RLHF). However, current reward models have limited generalization capabilities to unseen prompts and responses, which can lead to an unexpected phenomenon known as reward over-optimization, resulting in a decline in actual performance due to excessive optimization of rewards. While previous research has advocated for constraining policy optimization, our study introduces a novel approach to enhance the reward model's generalization ability against distribution shifts by regularizing the hidden states. Specifically, we retain the base model's language model head and incorporate a suite of text-generation losses to preserve the hidden states' text-generation capabilities, while concurrently learning a reward head behind the same hidden states. Our experimental results demonstrate that the introduced regularization technique markedly improves the accuracy of learned reward models across a variety of out-of-distribution (OOD) tasks and effectively alleviates the over-optimization issue in RLHF, offering a more reliable and robust preference learning paradigm [1].

## 1 Introduction

Pretrained large models have showcased impressive capabilities across diverse fields [1, 2, 3, 4, 5]. A notable trend in recent research is ensuring that large models align with human values and mitigate potentially harmful behaviors [6, 7, 8, 9, 10]. Alignment methods are crucial in achieving this objective, with two primary approaches being supervised fine-tuning (SFT) and reinforcement learning from human feedback (RLHF) [7, 8]. SFT directly finetunes the model using prompt and response pairs, proving to be a straightforward and efficient alignment technique [11, 12, 13]. Differently, RLHF begins by learning a reward model from user preferences and then employs reinforcement learning to optimize the language model to maximize rewards. A significant advantage of RLHF is its potential to generalize the reward model to unseen prompt-response pairs, effectively leveraging large volumes of unlabeled data [8, 14].

Despite the empirical success of RLHF, the challenge of training a reliable and generalizable reward model for unseen data remains an open problem. A well-known failure mode of reward model is known as "*overoptimization*" or "*reward hacking*" [15, 16, 17, 18], where policy optimization seemingly improves the proxy reward model but actually degrades the true reward function. [17] demonstrated in a synthetic setup that increasing the size of the reward model and the volume of training data can mitigate this overoptimization issue. However, such scaling is not always feasible

---

[1]Code and open-source reward models are available at https://github.com/YangRui2015/Generalizable-Reward-Model

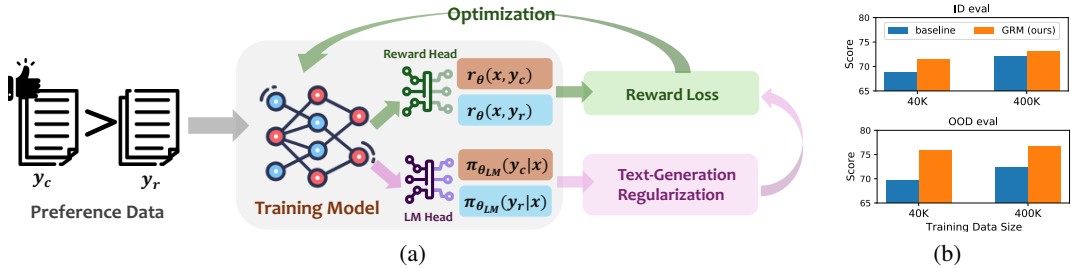

Figure 1: (1) Illustration of GRM. Given preference data pairs $(x, y_c, y_r)$, the reward head $r_\theta$ minimizes the reward loss in Eq 1, while the language model (LM) head $\pi_{\theta_{LM}}$ minimizes a suite of text-generation losses introduced in Sec 3.2. (2) Performance of GRM and the vanilla reward model on in-distribution (ID) task (Unified-Feedback) and average results of OOD tasks (HHH-Alignment and MT-Bench). Compared with the baseline reward model, GRM generalizes better on OOD tasks, with a larger advantage when the dataset size is relatively small.

in many realistic scenarios. To address this, a series of studies have been conducted, focusing either on enhancing the reward model with ensemble techniques [18, 19, 20, 21] or on constrained policy optimization [22, 23, 24, 25]. The latter paradigm is related to the offline RL literature [26, 27, 28, 29, 30, 31], which involves limiting the policy distribution to be close to the training data distribution. Among these, improving the generalization ability of reward models presents a fundamental and promising direction that can be studied independently from enhancements in policy optimization. Nevertheless, previous methods [18, 32] requiring training multiple reward models may be resource-intensive for the practical application of large models.

In this study, we present a lightweight yet effective solution designed to enhance the reward model's generalization ability against distribution shifts. Previous research [33] has theoretically shown that a randomly initialized head can distort pre-trained features, thereby negatively impacting out-of-distribution (OOD) performance. Inspired by this finding, we propose to regularize the feature during fine-tuning for preference learning using an adversarial regularizer, which derives a suite of text-generation losses. To this end, we introduce **Generalizable Reward Model (GRM)**, which retains the base model's language model head and regularizes the hidden states of the reward model by incorporating text-generation losses. This approach makes better use of the preference learning data while preserving the text generation capabilities of the hidden states. Notably, GRM does not necessitate training multiple reward models or relying on additional training data.

In our experiments, GRM substantially improves the evaluation accuracy of the reward model OOD evaluation datasets, demonstrating its superior ability to generalize learned preferences to unseen prompt and response pairs. Moreover, GRM consistently improves the performance of both 2B and 7B reward models, with a more pronounced improvement observed when the data size is limited. We also demonstrate that GRM can markedly enhance the performance of best-of-$n$ (BoN) sampling and PPO [34], effectively mitigating the overoptimization problem. These results highlight the potential of the GRM to serve as a more reliable proxy reward model for human preferences.

To conclude, our primary contributions are as follows:

- We propose GRM, a novel approach that employs text-generation regularization on the hidden states to enhance the generalization ability of reward models.
- Our study validates the effectiveness of all three types of text-generation regularization for GRM, identifying the SFT regularization as the most effective and stable solution.
- Our empirical results show that GRM significantly improves the accuracy of reward models across various OOD tasks. Furthermore, it consistently enhances the performance of RLHF, effectively alleviating the overoptimization problem.

## 2   Background

Typically, reinforcement Learning from Human Feedback (RLHF) involves reward modeling and policy optimization, with Best-of-$n$ Sampling (BoN) and Proximal Policy Optimization (PPO) being two commonly used methods for policy optimization.

**Reward Modeling.** Generally, reward modeling is based on the Bradley-Terry model [35], which aims to distinguish between the chosen response $y_c$ and the rejected response $y_r$ given the prompt $x$:

$$\mathcal{L}_{\text{reward}}(\theta) = -\mathbb{E}_{(x,y_c,y_r)\sim D}\left[\log\left(\sigma\left(r_\theta(x, y_c) - r_\theta(x, y_r)\right)\right)\right], \tag{1}$$

where $r_\theta(x, y)$ represents the reward score for prompt $x$ and output $y$ with model parameters $\theta$. $\sigma(\cdot)$ is the sigmoid function. By minimizing this loss function, the reward model assigns higher scores to outputs preferred by humans. Subsequently, the trained reward model can be used to guide the optimization of the language model.

**Best-of-$n$ Sampling (BoN).** BoN generates $n$ samples from the policy model, denoted as $Y_{\text{gen}}$, and then selects the best one based on scores provided by a reward model. BoN can be used for inference-time improvement or iterative optimization [36, 37, 38].

$$y_{\text{BON}}(x) = \arg\max_{y\in Y_{\text{gen}}} r_\theta(x, y). \tag{2}$$

**Proximal Policy Optimization (PPO).** PPO is a widely adopted method for RLHF in optimizing language models [16, 8, 39]. PPO learns a policy by maximizing a reward objective with a KL divergence penalty with coefficient $\eta$:

$$r_{\text{total}} = r_\theta(x, y) - \eta \text{KL}(\pi_{\text{PPO}}(y|x) \parallel \pi_{\text{SFT}}(y|x)), \tag{3}$$

where the KL penalty ensures that the optimized policy does not deviate significantly from the SFT policy to maintain the reliability of the reward model.

**Overoptimization.** Although the learned proxy reward model aims to approximate human preference, it may not consistently reflect authentic human preferences, potentially resulting in *over-optimization* [17, 18]. This issue emerges when the proxy reward model becomes overly optimized, causing the policy model to overfit certain erroneous patterns. Ultimately, this issue can diminish the model's alignment with actual human preferences, highlighting the need to ensure the reward model's robustness and reliability.

## 3 Method

In the common practice of training a reward model [8, 39, 40], reward models are initialized using a pretrained or SFT finetuned backbone, along with a randomly initialized reward head to predict the scores for prompt-response pairs. It's important to note that the backbone and original language model head are trained on a diverse range of datasets for text generation, which is distinct from the preference learning tasks. Under the task shift, the randomly initialized reward head can distort the pretrained features, thereby reducing the OOD generalization performance, as observed by [33]. We also confirm this impact on preference learning in Appendix C.1.

To improve the reward model's generalization capability against distribution shifts, we propose a lightweight yet effective solution, Generalizable Reward Model (GRM). This model employs a suite of text-generation regularizations for the hidden states. More specifically, GRM employs a structure as illustrated in Fig 1, with one language model (LM) head and one reward head sharing the same hidden states. The reward head is trained to minimize the reward loss $\mathcal{L}_{\text{reward}}$ in Eq 1, while the LM head is trained to maintain the text-generation ability of the hidden states during preference learning. Consequently, we define the overall loss function as follows:

$$\mathcal{L}_{\text{total}} = (1 - \alpha)\mathcal{L}_{\text{reward}} + \alpha\mathcal{L}_{\text{reg}}. \tag{4}$$

Here, $\alpha$ is the coefficient that balances the reward loss and the regularization. We will derive potential forms of the regularization term below.

### 3.1 Theoretical Motivation

To derive the potential formulation of the regularization term, we consider the following adversarial optimization problem: learning a reward model against an adversarial policy.

$$\theta = \arg\min_\theta\left\{\mathcal{L}_{\text{reward}}(\theta) + \gamma\max_\pi J(\theta, \pi)\right\}, \tag{5}$$

where $\gamma > 0$ is a coefficient. This objective is also considered by recent studies [24, 25] aiming to enhance DPO. Differently, we adopt it to learn a generalizable reward model.

The insight of Eq 5 is that we can enhance the robustness of the reward model by considering an adversarial policy $\pi$ from a certain policy class. The term for policy optimization $J(\theta, \pi)$ can have various formulations, but a KL divergence-regularized objective is generally used in training the policy [16, 8]. Moreover, it has an advantageous property that the inner optimization problem has an analytical solution, which can simplify the problem.

$$J(\theta, \pi) = \mathbb{E}_{x \sim D, y \sim \pi(\cdot|x)} \left[ r_\theta(x, y) \right] - \beta \mathbb{E}_{x \sim D} \left[ \text{KL} \left( \pi(\cdot|x) \parallel \pi_{\text{ref}}(\cdot|x) \right) \right], \tag{6}$$

where $\beta > 0$ is a regularization coefficient and $\pi_{\text{ref}}$ is the reference model. We denote the analytical solution of $J(\theta, \pi)$ as $\pi_\theta^*$. Incorporating $\pi_\theta^*$ into Eq 5, we can transform the min-max optimization problem into a standard optimization problem under certain assumptions:

$$\theta = \arg\min_\theta \{ (1 - \alpha)\mathcal{L}_{\text{reward}}(\theta) + \alpha_{\text{DPO}}\mathcal{L}_{\text{DPO}}(\pi_\theta^*) + \alpha_{\text{SFT}}\mathcal{L}_{\text{SFT}}(\pi_\theta^*) \} \tag{7}$$

Detailed derivation is deferred to Appendix A. Here, $\mathcal{L}_{\text{DPO}}$ is the same as the DPO objective [41] and $\mathcal{L}_{\text{SFT}}$ is the SFT objective that maximizes the probability of chosen responses. Notably, the two regularization terms originate from different sources: $\mathcal{L}_{\text{DPO}}$ stems from the reward loss, while $\mathcal{L}_{\text{SFT}}$ is derived from the adversarial term. This may explain why SFT regularization proves more beneficial than DPO regularization in our empirical results. Motivated by Eq 7, we relax the relationship between $r_\theta$ and $\pi_\theta^*$ and propose learning a reward model parameterized by $\theta$ and a language model head parameterized by $\theta_{\text{LM}}$, both sharing the same hidden states. A discussion of this design can be found in Appendix A. Below, we detail three practical implementations.

### 3.2 Text-Generation Regularization

Inspired by Eq 7, we train the LM head to minimize text-generation losses, such as DPO and SFT losses, as the regularization term for GRM. To independently study the effectiveness of these two regularizations and reduce GPU memory usage, we introduce three practical implementations: DPO regularization, DPO without reference regularization, and SFT regularization.

**DPO Regularization.** By setting $\alpha_{\text{DPO}} = \alpha$ and $\alpha_{\text{SFT}} = 0$ in Eq 7, we can directly adopt the DPO loss as a regularization term for GRM to regularize the hidden states:

$$\mathcal{L}_{\text{DPO}}(\theta_{\text{LM}}) = -\mathbb{E}_{(x, y_c, y_r) \sim D} \left[ \log \sigma \left( \beta \log \left( \frac{\pi_{\theta_{\text{LM}}}(y_c \mid x)}{\pi_{\text{ref}}(y_c \mid x)} \right) - \beta \log \left( \frac{\pi_{\theta_{\text{LM}}}(y_r \mid x)}{\pi_{\text{ref}}(y_r \mid x)} \right) \right) \right], \quad (8)$$

where $\pi_{\text{ref}}$ is the base model serving as the reference model, and $\pi_{\theta_{\text{LM}}}$ is our optimized policy. $\beta$ is a coefficient that controls the KL penalty between $\pi_{\theta_{\text{LM}}}$ and $\pi_{\text{ref}}$. Notably, $\pi_{\theta_{\text{LM}}}$ shares the same base model with the reward model $r_\theta$, except for the output layer.

**DPO Regularization w/o Reference Model.** While straightforward, the use of a reference model in DPO regularization can be memory-intensive for large models. To address this, and inspired by prior works that eliminate the need for reference model [42, 43], we introduce the DPO regularization without a reference model, denoted as $\mathcal{L}_{\text{DPO-noref}}$. This method reduces the need for large GPU memory during training. The loss function $\mathcal{L}_{\text{DPO-noref}}$ is defined as:

$$\mathcal{L}_{\text{DPO-noref}}(\theta_{\text{LM}}) = -\mathbb{E}_{(x, y_c, y_r) \sim D} \left[ \log \sigma \left( \beta \log \left( \frac{\pi_{\theta_{\text{LM}}}(y_c \mid x)}{\pi_{\theta_{\text{LM}}}(y_r \mid x)} \right) \right) \right]. \tag{9}$$

**SFT Regularization.** By setting $\alpha_{\text{DPO}} = 0$ and $\alpha_{\text{SFT}} = \alpha$ in Eq 7, we can simplify the regularization term to SFT regularization, thereby reducing the computational cost. This method only maximizes the probability of the chosen responses:

$$\mathcal{L}_{\text{SFT}}(\theta_{\text{LM}}) = -\mathbb{E}_{(x, y_c) \sim D} \left[ \log \sigma \left( \beta \log \left( \pi_{\theta_{\text{LM}}}(y_c \mid x) \right) \right) \right]. \tag{10}$$

This equation differs slightly from the standard SFT objective to maintain coherence with the above two cases within the regularization suite and avoid the need for hyperparameter adjustments for $\alpha$. Please refer to Appendix C.3 for a discussion.

### 3.3 Advantages of GRM

In summary, GRM offers three key advantages: **(1) Mitigating feature distortion.** The application of text-generation loss helps maintain the text-generation ability of the base model and prevents

excessive feature distortion. Simultaneously, it also adapts the model to the data distribution of preference learning. **(2) Prevention of Overfitting.** The text-generation regularization derived from an adversarial training objective helps prevent the reward model from overfitting to certain spurious features, which can be detrimental to OOD generalization. This effect becomes more pronounced when the preference data includes erroneous comparison pairs or when the dataset size is limited. **(3) Efficiency.** GRM is an efficient solution that does not require training multiple reward models or additional training data. Additionally, different choices of loss type entail varying memory and computational costs. Interestingly, we find that the simplest option, SFT regularization, proves to be the most stable choice.

## 4   Experimental Setup

**Datasets.** For training reward models, we leverage the Unified-Feedback dataset [2], which stands as one of the largest collections of pairwise feedback datasets. In Section 5.1, we train all reward models on a subset of 400K and 40K samples from the Unified-Feedback dataset and evaluate them on the hold-out 8K eval set. In addition, for evaluating model performance on out-of-distribution (OOD) preference data, we utilize datasets such as HHH-Alignment [3] [44], MT-Bench Human Judgements [4] [45], and RewardBench [46]. The HHH-Alignment dataset evaluates language models on helpfulness, honesty, and harmlessness, while the MT-Bench dataset contains human preferences for model responses to MT-bench questions. Besides, RewardBench is a new benchmark designed to evaluate the capabilities and safety of reward models. We consider HHH-Alignment, MT-Bench, and RewardBench as OOD evaluation tasks because the prompt and response distributions differ from our training distribution. For the RLHF experiments in Section 5.2 we downsample 20K data from Unified-Feedback for training reward models and optimizing the PPO policy, and another 1K data for evaluating BoN or the learned PPO policy.

**Base Models.** In the preference learning experiments, our base models include gemma-2B-it [47] and Mistral-7B-Instruct-v0.2 [48]. For the RLHF experiments, gemma-2B-it serves as the policy model for both BoN and PPO experiments, whereas the gold reward model [5] is a 7B human preference model finetuned using the entire Unified-Feedback dataset.

**Baselines.** We compare the performance of GRM with several baselines, including *Baseline Classifier* trained using the original reward loss in Eq 1; *Frozen Classifier* that fixes the base model's feature and only finetunes a nonlinear classification head; *Margin* that adds an additional margin in the original reward loss [10, 39]; *Label Smooth* that mitigate the overfitting problem by penalizing overconfident model outputs [39]; *Ensemble* method with a group of 3 reward models [18] to calculate the average or minimum values as rewards. In addition, for RewardBench, we present the performance of several existing open-source state-of-the-art reward models for better reference, including PairRM [49], Starling-RM-7B/34B [50], and UltraRM-13B [51]. For more experimental details and additional results, please refer to Appendix B and Appendix C, respectively.

## 5   Evaluation Results

We present a comprehensive evaluation of GRM, utilizing both in-distribution (ID) and out-of-distribution (OOD) datasets, as well as existing benchmarks for reward models. Furthermore, we explore the impact of GRM on the overoptimization issue in RLHF. Our primary findings can be summarized as follows:

- GRM significantly enhances the generalization capability of reward models, resulting in substantial improvements on both ID and various OOD evaluation sets (Section 5.1).
- All three types of text-generation regularization losses can improve the generalization performance, with the SFT regularization being the most effective and stable (Section 5.1).
- GRM demonstrates robustness in the limited dataset setting, outperforming baselines by an even larger margin (Section 5.1).

---

[2]https://huggingface.co/datasets/llm-blender/Unified-Feedback
[3]https://huggingface.co/datasets/HuggingFaceH4/hhh_alignment
[4]https://huggingface.co/datasets/lmsys/mt_bench_human_judgments
[5]reward-model-Mistral-7B-instruct-Unified-Feedback

- GRM effectively mitigates the overoptimization issue in both BoN and PPO (Section 5.2).

- GRM also exhibits robustness against label noise in the preference dataset (Section 5.2).

Table 1: Results on ID and OOD evaluation with **400K training data** from Unified-Feedback. The best performance in each task is in bold and the second best one is underlined.

Table 2: Results on ID and OOD evaluation with **40K training data** from Unified-Feedback. The best performance in each task is in bold and the second best one is underlined.

| Reward Model | Unified Feedback | HHH Alignment | MT Bench | Reward Model | Unified Feedback | HHH Alignment | MT Bench |
|---|---|---|---|---|---|---|---|
| Classifier (Frozen) | 63.8 | 66.4 | 69.5 | Classifier (Frozen) | 63.9 | 68.6 | 68.2 |
| Classifier (baseline) | 72.1 | 73.4 | 71.2 | Classifier (baseline) | 68.8 | 70.3 | 69.1 |
| Classifier + margin | 72.0 | 75.0 | 72.6 | Classifier + margin | 69.6 | 69.8 | 71.0 |
| Classifier + label smooth | 71.5 | 72.1 | 71.2 | Classifier + label smooth | 68.5 | 68.8 | 71.9 |
| Classifier + Ensemble | 72.8 | 76.8 | **73.7** | Classifier + Ensemble | 69.9 | 72.2 | 71.1 |
| **GRM w/ dpo (ours)** | 73.8 | 79.2 | 73.4 | **GRM w/ dpo (ours)** | 70.2 | 71.6 | 71.3 |
| **GRM w/ dpo-noref (ours)** | **73.9** | 79.7 | 73.0 | **GRM w/ dpo-noref (ours)** | 71.4 | 76.6 | 72.1 |
| **GRM w/ sft (ours)** | 73.2 | **79.8** | 73.4 | **GRM w/ sft (ours)** | **71.5** | **78.7** | **73.0** |

## 5.1 Evaluation on Reward Modeling

**ID and OOD Evaluation.** The results, shown in Table 1 and Table 2, illustrate the evaluation performance of different reward modeling methods using the gemma-2B-it base model on both ID (Unified-Feedback) and OOD (HHH-Alignment and MT-Bench) datasets. Regardless of the size of the training data (400K or 40K), our proposed method, GRM, with three types of regularizations, consistently outperforms the baseline models on both the ID evaluation set and the two OOD datasets. For instance, GRM w/ sft with 400K training data enhances the baseline from 72.1 to 73.2 in ID score, and improves the HHH-Alignment score from 73.4 to 79.8 and the MT-Bench score from 71.2 to 73.4. Notably, the improvement in OOD performance is significantly larger than that in ID. These results suggest that the GRM methods are highly effective in evaluating unseen preference data, demonstrating substantially robust generalization capabilities.

Regarding other baseline models, the Frozen classifier, which maintains its base model's parameters, exhibits the lowest ID and OOD scores. This suggests that the pretrained features of the base model alone are insufficient for effective preference learning, emphasizing the importance of fine-tuning the base model's features to the preference task. Furthermore, the margin loss and label smoothing techniques do not consistently improve the ID and OOD tasks, whereas the ensemble baseline consistently enhances both ID and OOD scores. Despite requiring the training of multiple reward models, ensemble-based methods still do not surpass GRM, particularly when learning from a 40K training set. These results highlight the substantial improvement and generalization capability of GRM in preference learning.

**Comparison of Different Regularizations.** As observed in Table 1, when the training dataset is sufficiently large, GRM with three types of regularizations (namely GRM w/ dpo, GRM w/ dpo-noref, and GRM w/ sft) perform comparably. This demonstrates that GRM is robust to the choice of regularization type when the dataset is large. However, in Table 2, where the training data is limited to 40K, a clear trend emerges: GRM w/ sft outperforms GRM w/ dpo-noref, which in turn outperforms GRM w/ dpo, on both the ID and OOD scores. Interestingly, the simplest form of regularization, SFT regularization, not only requires the lowest training cost but also yields the most stable overall results. Consequently, we adopt it as the default choice for our subsequent study.

**Results on RewardBench.** In Table 3 and Table 4, we evaluate GRM and various baselines on RewardBench across chat, chat-hard, safety, and reasoning task groups. We consider a variant of GRM with a linear reward head instead of the default nonlinear reward head as detailed in Appendix B. In Table 3, the 7B baseline matches the score of Starling-RM-7B [50], while GRM (linear) w/ sft demonstrates a considerable improvement, increasing the average score from 76.3 to 79.5. Comparing variants of GRM, we can conclude that: (1) SFT regularization performs better than the DPO w/o reference model regularization, and (2) GRM with a linear head achieves a better overall score than that with a nonlinear head, especially in the challenging reasoning task group.

Table 3: Results on RewardBench with **400K training data** from Unified-Feedback.

| Reward model | Average | Chat | Chat-Hard | Safety | Reasoning |
|---|---|---|---|---|---|
| PairRM | 58.7 | 90.2 | 53.0 | 31.5 | 60.0 |
| Starling-RM-7B | 76.2 | 98.0 | 43.4 | 88.6 | 74.6 |
| Starling-RM-34B | **84.0** | 96.9 | 59.0 | 89.9 | 90.3 |
| UltraRM-13B | 69.8 | 96.1 | 55.3 | 45.8 | 82.0 |
| *Base Model: Gemma 2b it* | | | | | |
| Classifier (baseline) | 68.2 | 95.5 | 38.0 | 73.8 | 65.3 |
| Classifier + margin | 70.2 | 95.8 | 38.4 | 73.9 | 72.5 |
| Classifier + label smooth | 70.6 | 94.4 | 37.3 | 73.2 | **77.4** |
| Classifier + Ensemble | 71.0 | **98.0** | 37.5 | 77.3 | 71.3 |
| GRM (linear) w/ dpo noref (ours) | 70.2 | 96.7 | 39.0 | 76.4 | 68.5 |
| GRM (linear) w/ sft (ours) | **71.5** | 96.1 | 40.1 | **80.3** | 69.3 |
| GRM w/ dpo noref (ours) | 70.2 | 95.8 | 40.1 | 78.7 | 66.2 |
| GRM w/ sft (ours) | 70.8 | 97.8 | **42.1** | 77.9 | 65.2 |
| *Base Model: Mistral 7b Instruct* | | | | | |
| Classifier (baseline) | 76.3 | 96.6 | 52.4 | 86.7 | 69.5 |
| Classifier + margin | 74.5 | 96.4 | 51.5 | 85.3 | 64.8 |
| Classifier + label smooth | 76.3 | 97.2 | 49.8 | 85.8 | 72.3 |
| Classifier + Ensemble | 76.6 | 96.6 | 51.8 | 85.1 | 73.0 |
| GRM (linear) w/ dpo noref (ours) | 78.3 | **98.0** | 53.3 | **86.4** | 75.3 |
| GRM (linear) w/ sft (ours) | **79.5** | 97.8 | 54.6 | 86.3 | **79.2** |
| GRM w/ dpo noref (ours) | 78.0 | 97.8 | 54.0 | 85.7 | 74.4 |
| GRM w/ sft (ours) | 77.6 | **98.0** | **55.3** | 85.8 | 71.2 |

Table 4: Results on RewardBench with **40K training data** from Unified-Feedback.

| Reward model | Average | Chat | Chat-Hard | Safety | Reasoning |
|---|---|---|---|---|---|
| *Base Model: Gemma 2B it* | | | | | |
| Classifier (baseline) | 64.5 | 95.8 | 37.3 | 59.9 | 64.8 |
| Classifier + margin | 66.1 | **97.2** | 37.5 | 56.8 | 72.7 |
| Classifier + label smooth | 61.1 | 91.6 | 39.0 | 53.8 | 60.2 |
| Classifier + Ensemble | 65.2 | 96.1 | 38.2 | 58.8 | 67.6 |
| GRM (linear) w/ dpo noref (ours) | 61.7 | 94.7 | 38.4 | 62.5 | 51.2 |
| GRM (linear) w/ sft (ours) | **69.5** | 94.7 | 40.8 | 65.4 | **77.0** |
| GRM w/ dpo noref (ours) | 66.6 | 92.5 | 39.9 | **72.5** | 61.4 |
| GRM w/ sft (ours) | 66.8 | 94.1 | **41.9** | 69.5 | 61.5 |
| *Base Model: Mistral 7B Instruct* | | | | | |
| Classifier (baseline) | 68.2 | 89.7 | 50.7 | 74.7 | 57.9 |
| Classifier + margin | 62.8 | 89.7 | 47.1 | 70.7 | 43.6 |
| Classifier + label smooth | 72.1 | 94.1 | 47.1 | 67.5 | 79.7 |
| Classifier + Ensemble | 69.3 | 89.6 | 50.2 | 72.7 | 59.0 |
| GRM (linear) w/ dpo noref (ours) | 77.8 | 96.9 | 52.9 | 82.7 | 78.8 |
| GRM (linear) w/ sft (ours) | 78.3 | 96.7 | 52.4 | 81.5 | **82.5** |
| GRM w/ dpo noref (ours) | **78.6** | **97.8** | **54.6** | 82.0 | 79.9 |
| GRM w/ sft (ours) | 78.4 | 97.2 | 54.2 | **83.6** | 78.6 |

Regarding the baselines, consistent with previous results, the margin loss and label smoothing do not provide a coherent improvement over the baseline. While ensemble methods effectively improve upon the baseline, they still underperform GRM. Overall, these results demonstrate that GRM is a strong contender in reward modeling tasks, exhibiting superior performance across various benchmarks.

**Comparison of Different Dataset Sizes.** Another noteworthy observation is that GRM exhibits greater robustness to the size of the training dataset compared to baselines. For instance, in Table 1 and Table 2, when the training data size decreases from 400K to 40K, the baseline's HHH Alignment score and MT-Bench score drop from 73.4 and 71.2 to 70.3 and 69.1, respectively. In contrast, GRM with SFT regularization only slightly drops from 79.8 and 73.4 to 78.7 and 73.0, respectively. This

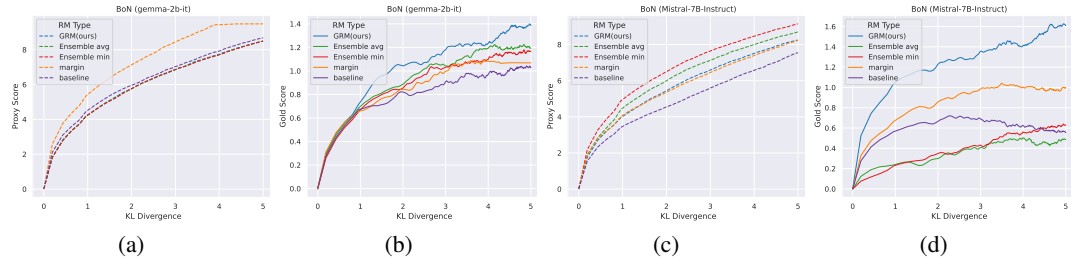

Figure 2: Proxy scores and gold scores of BoN experiments for base models of (a)(b) gemma-2b-it and (c)(d) Mistral-7B-Instruct. Proxy and gold scores are in dashed and solid curves, respectively. Rewards are normalized to start from 0. GRM demonstrates a robust ability to select the best response aligned with the gold rewards as the KL Divergence increases.

trend is consistent in Table 3 and Table 4. Specifically, for the Mistral 7B Instruct base model, the baseline's average score drops from 76.3 to 68.2 when learning from 40K training data, while GRM (linear) w/ sft only drops from 79.5 to 78.3. These findings suggest that the prior reward training paradigms are sensitive to smaller dataset sizes. In contrast, GRM is robust even with a limited dataset.

**Full Parameter Training Results on a Larger Dataset.**    To further demonstrate the effectiveness of GRM, we trained the GRM using the llama3-8b-instruct model [52]. We perform a full parameter fine-tuning for 1 epoch on one of the largest open-source preference datasets [6]. Our results, presented in Table 5, highlight the considerable potential of scaling GRM to larger models and datasets. Especially, GRM outperforms a 34B reward model and even GPT-4 as a judge. It is worth noting that the GRM significantly improves the performance of the 8B reward model from 84.7 to 87.0, using the same base model and training data as FsfairX-LLaMA3-RM-v0.1 [40]. This improvement is particularly remarkable in the challenging 'Reasoning' section.

Table 5: Results of full parameter training on RewardBench.

| Reward model | Average | Chat | Chat-Hard | Safety | Reasoning |
|---|---|---|---|---|---|
| GRM (Ours, 8B) | **87.0** | 98.6 | 67.8 | **89.4** | **92.3** |
| gpt-4-0125-preview | 85.9 | 95.3 | 74.3 | 87.2 | 86.9 |
| gpt-4-turbo-2024-04-09 | 85.1 | 95.3 | **75.4** | 87.1 | 82.7 |
| FsfairX-LLaMA3-RM-8B | 84.7 | **99.4** | 65.1 | 87.8 | 86.4 |
| Starling-RM-34B | 82.7 | 96.9 | 57.2 | 88.2 | 88.5 |

## 5.2    Evaluation on RLHF

**Best-of-$n$ Sampling (BoN).**    Fig 2 presents the results of BoN sampling for base models of sizes 2B and 7B. For all BoN experiments, we utilize a 20K subset from the Unified-Feedback dataset, labeled by the gold reward model, to train proxy reward models. Following the [17, 18], we conduct BoN sampling on a 1K held-out test set from $n$ responses for each prompt, based on the scores of the trained proxy model. The selected responses are then scored using the gold reward model, and their gold scores are averaged over the 1K test prompts. The average gold score reveals the true quality of the responses selected by the proxy reward models. We set the KL Divergence from 0 to 5, corresponding to the number of responses $n$ ranging from 1 to 405 for each prompt, according to the equation $\text{KL}_{\text{BoN}} = \log n - \frac{n-1}{n}$. Ideally, a good proxy reward model should yield larger average proxy and gold scores as the KL increases. However, the average gold scores of some baseline methods plateau or even drop after KL > 4, such as the baseline reward model in Fig 2(d), despite their proxy scores continuing to increase in Fig 2(c). This suggests that these reward models suffer from the overoptimization issue.

---

[6]https://huggingface.co/datasets/hendrydong/preference_700K

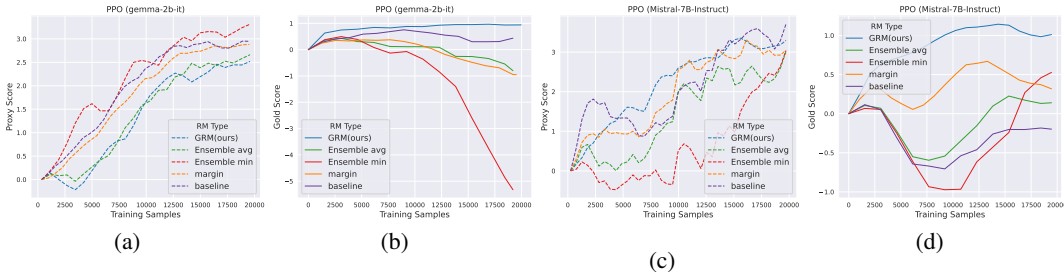

Figure 3: Proxy scores and gold scores of PPO experiments for reward model based on (a)(b) gemma-2b-it and (c)(d) Mistral-7B-Instruct. All rewards are normalized to start from 0.

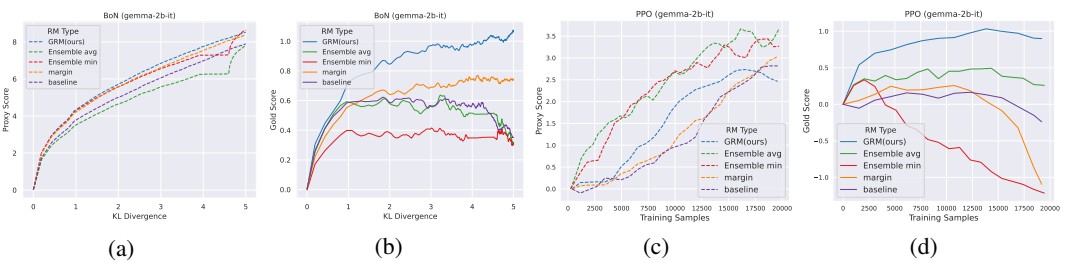

Figure 4: Proxy scores and gold scores of (a)(b) BoN experiments and (c)(d) PPO experiments with 25% **label noise**. All rewards are normalized to start from 0.

In contrast, GRM consistently demonstrates an increase in both the proxy score and gold score, indicating that it effectively mitigates over-optimization. This advantage is more pronounced in the 7B base model, where GRM achieves an average gold score of 1.5, while the baseline reward model only reaches a score of 0.5. Regarding other baselines, we find that the margin loss and ensemble methods (especially the 'min' strategy) contribute to the robustness of the reward model. However, they still do not compare favorably with GRM. These results underscore the strong potential of GRM to serve as a reliable and robust proxy reward model for RLHF.

**Proximal Policy Optimization (PPO).**   To investigate whether GRM can effectively mitigate the overoptimization issue in PPO, we further employ those 2B and 7B reward models obtained from the BoN experiments to fine-tune the policy model (gemma-2b-it) using PPO. The training and evaluation datasets are identical to the BoN experiments. We train PPO for one epoch on the training set, comprising 20K training samples. As depicted in Fig 3, PPO exhibits a stronger tendency to hack the learned reward models compared to BoN. The gold scores of baseline methods begin to decline early in the training process, while their proxy scores increase, indicating a clear overoptimization issue. In contrast, GRM demonstrates superior robustness in terms of the gold score, which rises with the increase in proxy scores. This validates that GRM can effectively alleviate overoptimization for PPO. Please refer to Appendix D for a clear comparison of the results generated by PPO.

**Robustness to Label Noise.**   Human preference data typically contains around 20 to 30% noise, as highlighted in previous studies [39]. Such inconsistent preference data can render the reward model less generalizable [32, 53] and hinder policy learning [54, 55, 56], leading to a performance decline. To evaluate the robustness of GRM against label noise, we incorporate a 25% label noise into the 20K training data for all proxy reward models. The results are depicted in Fig 4. Most gold scores expose a more severe over-optimization issue, as compared to the results in Fig 2(b) and Fig 3(b), indicating that those reward models are heavily overfitting under the noisy label setting. On the contrary, GRM exhibits superior robustness under noisy conditions, consistently achieving a peak gold score over 1.0 without a significant decline. This demonstrates that GRM is highly accurate and robust at measuring the sample quality, even in the presence of noise within the training data.

# 6   Related Works

**Reward Modeling.** Reward models, trained on human preference data, are crucial in guiding RLHF [8, 57] or prompt optimization [58]. Recent studies have concentrated on developing advanced reward models to improve the performance of LLMs in RLHF. One approach involves enhancing reward modeling by improving the quality or quantity of preference data [59, 60, 61]. Other strategies focus on learning token-wise dense rewards [62, 63] or multi-objective rewards [38]. Additionally, a series of works aim to enhance the robustness of reward models against preference inconsistencies. Techniques such as adaptive margin [10], contrastive learning [39], and meta-learning [64] are employed to improve the model's ability to differentiate between chosen and rejected responses.

**Mitigating Overoptimization in RLHF.** Reward models tend to overfit and struggle to generalize beyond the training distribution, which often leads to the overoptimization issue [17]. One approach to mitigate this is to penalize overly confident model outputs using label smoothing [39] or SFT regularization [24, 25]. Alternatively, the model and data can be iteratively updated, replacing hard labels with soft labels [65]. Ensemble techniques, which train several reward models, can also help reduce reward hacking and manage shifts in distribution [18, 19, 20, 66, 32, 67, 68]. Adversarial Preference Optimization employs adversarial learning between reward models and an LLM agent to address the gap in generation distribution [69]. Recent studies have also utilized uncertainty to mitigate reward over-optimization, including the integration of an uncertainty penalty into rewards [70], or the construction of a confidence interval for gold rewards based on uncertainty estimations [23].

# 7   Conclusion

In this study, we introduce an efficient approach aimed at enhancing the generalizability and robustness of reward learning for large language models. By incorporating regularization techniques on the hidden states of reward models, our method demonstrates substantial improvements in the generalization performance of reward models for unseen data. Moreover, our approach effectively mitigates the issue of overoptimization in RLHF. We believe that our findings hold promise in inspiring future research efforts towards the development of more robust reward models that can facilitate the alignment of large models through cost-effective solutions.

## Limitations

In this study, we evaluate the robustness of GRM against label noise by introducing a 25% level of synthetic noise into the training data for all proxy reward models. This is achieved by randomly flipping chosen and rejected labels. Due to cost considerations, we conduct synthetic experiments in line with community practices [18, 39], as using human-labeled data is not feasible for us. However, synthetic data may introduce biases that don't accurately reflect real-world scenarios. Future research should aim to mitigate this limitation by incorporating experiments with human-labeled data, providing a more thorough evaluation of the reward model's robustness. Another limitation of our study is the computational restriction preventing us from testing GRM with parameter sizes exceeding 10B. Further efforts to extend our method to larger reward models could be highly promising.

## Acknowledgement

Tong Zhang is partially supported by an NSF IIS grant No. 2416897. Huan Zhang was supported by the AI2050 program at Schmidt Sciences (AI 2050 Early Career Fellowship). The authors would like to thank the reviewers and readers for constructive feedback on the manuscript.

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

# A Deriving the Regularization Term

To derive the potential formulation of the regularization term, we consider the following adversarial optimization problem: learning a reward model against an adversarial policy.

$$\theta = \arg\min_{\theta} \left\{ \mathcal{L}_{\text{reward}}(\theta) + \gamma \max_{\pi} J(\theta, \pi) \right\} \tag{11}$$

The term for policy optimization $J(\theta, \pi)$ can have different formulations, but a KL divergence regularized optimization objective is generally used in training the policy [16, 8, 71]. Moreover, it has an advantageous property that the inner optimization problem has an analytical solution, which can simplify the problem.

$$J(\theta, \pi) = \mathbb{E}_{x \sim D, y \sim \pi(\cdot|x)} \left[ r_\theta(x, y) \right] - \beta \mathbb{E}_{x \sim D} \left[ \text{KL} \left( \pi(\cdot|x) | \pi_{\text{ref}}(\cdot|x) \right) \right], \tag{12}$$

where $\beta > 0$ is the coefficient controlling the regularization degree and $\pi_{\text{ref}}$ is the reference model. The analytical solution of $J(\theta, \pi)$ is formulated as follows:

$$\pi_\theta^* = \frac{1}{Z_\theta(x)} \pi_{\text{ref}}(y|x) \exp\left( r_\theta(x, y)/\beta \right), Z_\theta(x) = \sum_{y'} \pi_{\text{ref}}(y'|x) \exp\left( r_\theta(x, y')/\beta \right) \tag{13}$$

Equivalently, we can obtain the formulation of reward described by $\pi_\theta^*$ and $\pi_{\text{ref}}$ as in [41]:

$$r_\theta(x, y) = \beta \left( \log \pi_\theta^*(y|x) - \log \pi_{\text{ref}}(y|x) + \log Z_\theta(x) \right) \tag{14}$$

Following recent theoretical analysis [25], we define a fixed calibration policy $\pi_{\text{cal}}$ that is independent of the algorithm, which has the calibration effect of centering the reward function while incorporating additional policy preferences into the objective.

**Definition 1** $\pi_{\text{cal}}$ *is a fixed calibration policy for reward model $r_\theta$ and the dataset $D$ that satisfies:*

$$\mathbb{E}_{x \sim D, y \sim \pi_{\text{cal}}}[r_\theta(x, y)] = 0$$

Therefore, we can rewrite $\max_\pi J(\theta, \pi)$ as:

$$\begin{aligned}
\max_\pi J(\theta, \pi) = J(\theta, \pi_\theta^*) &= \mathbb{E}_{x \sim D, y \sim \pi_\theta^*(\cdot|x)} \left[ r_\theta(x, y) - \beta(\log \pi_\theta^*(y|x) - \log \pi_{\text{ref}}(y|x)) \right] \\
&= \mathbb{E}_{x \sim D, y \sim \pi_\theta^*(\cdot|x)} \left[ \log Z_\theta(x) \right] = \mathbb{E}_{x \sim D, y \sim \pi_{\text{cal}}(\cdot|x)} \left[ \log Z_\theta(x) \right] \\
&= \mathbb{E}_{x \sim D, y \sim \pi_{\text{cal}}(\cdot|x)} \left[ r_\theta(x, y) - \beta(\log \pi_\theta^*(y|x) - \log \pi_{\text{ref}}(y|x)) \right] \\
&= -\beta \mathbb{E}_{x \sim D, y \sim \pi_{\text{cal}}(\cdot|x)} \left[ \log \pi_\theta^*(y|x) - \log \pi_{\text{ref}}(y|x) \right].
\end{aligned} \tag{15}$$

The second line is established because $\log Z_\theta(x)$ is independent of the distribution $y$. Besides, the last line just adopts the definition of $\pi_{\text{cal}}$.

Incorporating Eq 15 and Eq 14 into Eq 11, we can transform the min-max optimization problem into a standard optimization problem by considering the policy $\pi_\theta^*$:

$$\begin{aligned}
\theta &= \arg\min_{\theta} \left\{ (1-\alpha)\mathcal{L}_{\text{reward}}(\theta) + \alpha\mathcal{L}_{\text{reward}}(\theta) + \gamma \max_{\pi} J(\theta, \pi) \right\} \\
&= \arg\min_{\theta} \left\{ (1-\alpha)\mathcal{L}_{\text{reward}}(\theta) - \alpha \mathbb{E}_{(x, y_c, y_r) \sim D} \log \sigma \left( \beta \log \left( \frac{\pi_\theta^*(y_c \mid x)}{\pi_{\text{ref}}(y_c \mid x)} \right) - \beta \log \left( \frac{\pi_\theta^*(y_r \mid x)}{\pi_{\text{ref}}(y_r \mid x)} \right) \right) \right. \\
&\quad \left. - \gamma\beta \mathbb{E}_{x \sim D, y \sim \pi_{\text{cal}}(\cdot|x)} \left[ \log \pi_\theta^*(y|x) - \log \pi_{\text{ref}}(y|x) \right] \right\} \\
&= \arg\min_{\theta} \{ (1-\alpha)\mathcal{L}_{\text{reward}}(\theta) + \alpha\mathcal{L}_{\text{DPO}}(\pi_\theta^*) - \gamma\beta \mathbb{E}_{x \sim D, y \sim \pi_{\text{cal}}(\cdot|x)} \left[ \log \pi_\theta^*(y|x) \right] \}
\end{aligned} \tag{16}$$

Here, we use $\mathcal{L}_{\text{DPO}}(\pi_\theta^*)$ to replace the second term, as it is the same as DPO objective [41]. In the second line, we put the reward described by Eq 14 into $\alpha\mathcal{L}_{\text{reward}}$. In the final step, we remove the $\pi_{\text{ref}}$ term as it does not depend on the parameters of the reward $r_\theta$, unlike $\pi_\theta^*$ which is dependent on reward $r_\theta$.

Interestingly, if we set the calibration policy $\pi_{\text{cal}}$ as the chosen responses $y_c$ from the dataset $D$, the last term becomes an SFT loss. Thus, we can derive the general regularization terms in our framework by renaming the coefficients for $\pi_\theta^*$ as $\alpha_{\text{DPO}}$ and $\alpha_{\text{SFT}}$, and removing the constraint that $\alpha_{\text{DPO}} = \alpha$.

$$\arg\min_{\theta} \{ (1-\alpha)\mathcal{L}_{\text{reward}}(\theta) + \alpha_{\text{DPO}}\mathcal{L}_{\text{DPO}}(\pi_\theta^*) + \alpha_{\text{SFT}}\mathcal{L}_{\text{SFT}}(\pi_\theta^*) \} \tag{17}$$

Notably, the two regularization terms come from different sources, where $\mathcal{L}_{\text{DPO}}$ is from the reward loss and $\mathcal{L}_{\text{SFT}}$ is derived from the adversarial term. This may be the reason why SFT regularization is more helpful than DPO regularization in our empirical results. Inspired by the objective in Eq 7, we relax the relationship between $r_\theta$ and $\pi_\theta^*$ and propose to learn a reward model parameterized by $\theta$ and a language model head parameterized by $\theta_{\text{LM}}$, both sharing the same hidden states.

**Discussion.** In Eq 17, we retain both the reward model $r_\theta$ and the policy $\pi_\theta^*$, and replace $\pi_\theta^*$ with a language head $\pi_{\theta_{\text{LM}}}$. A simpler solution is to keep only the reward model by replacing $\pi_\theta^*$ with $r_\theta$, which leads to the following objective:

$$\arg\min_\theta \{\mathcal{L}_{\text{reward}}(\theta) - \gamma\mathbb{E}_{x,y_c \sim D}[r_\theta(x, y_c)] + \gamma\beta\mathbb{E}_{x \sim D}[\log Z_\theta(x)]\}.$$

This approach can be understood as minimizing reward loss while applying regularization to maximize the rewards of selected responses relative to the overall rewards. However, this method is limited by the inefficient calculation of $Z_\theta$ over the response distribution generated by $\pi_{\text{SFT}}$. Therefore, we propose our solution, GRM, which involves a reward model that shares hidden states with a language head. This setup captures certain correlations through shared parameters, helps prevent feature distortion, and is both cost-effective and highly efficient.

## B  Implementation Details

**Baseline Details.** All baseline reward models employ the "AutoModelForSequenceClassification" class from transformers [72], which utilizes a randomly initialized linear head to derive rewards. We then train each reward model to minimize the loss function with the training data. For **ensemble baselines**, we train 3 reward models with different random seeds and aggregate their outputs via the 'average' or the 'minimum' strategy. We adopt the average value for the ensemble baseline in Section 5.1 as we find that the minimum value can decrease accuracy and underperform the average one. But for the RLHF experiments in Section 5.2, we report both results because we find some sometimes the 'minimum' strategy can work better than due to its pessimism.

The **margin loss** function [10] is defined as below:

$$\mathcal{L}_{\text{margin}}(\theta) = -\mathbb{E}_{(x,y_c,y_r) \sim D}\left[\log\left(\sigma\left(r_\theta(x, y_c) - r_\theta(x, y_r) - m(r)\right)\right)\right],$$

which enhances the reward model by emphasizing the differences in rewards. We use the scores between chosen and rejected responses in the Unified-Feedback dataset to calculate $m(r)$.

Additionally, the **label smooth loss** is defined as

$$\mathcal{L}_{\text{smooth}}(\theta) = -\mathbb{E}_{(x,y_c,y_r) \sim D}\left[(1 - \epsilon)\log\left(\sigma\left(r_\theta(x, y_c) - r_\theta(x, y_r)\right)\right) - \epsilon\log\left(\sigma\left(r_\theta(x, y_c) - r_\theta(x, y_r)\right)\right)\right],$$

where we set $\epsilon = 0.1$. The label smooth loss function enhances the model's resilience to a certain degree of errors, thereby alleviating the problem of overfitting.

**GRM Details.** For GRM, the default reward head is configured as a linear layer with shape (hidden size, 1024), followed by a ReLU activation function, and another linear layer of shape (1024, 1). The weight of the text-generation regularization $\alpha$ is set to 0.01 and the coefficient $\beta$ in our regularizations is set to 0.1 by default. In the case of the GRM (linear) variant, the reward head is directly set as a linear layer of shape (hidden size, 1). We found a smaller $\alpha = 0.001$ is better for the linear variant.

**Training and Evaluation Details.** We implement all methods based on transformers [72] and trl [73]. More details are listed in Table 6. To use the Unified-Feedback dataset, we downsample the training data from the 'all' set and use all the 8K test data for evaluation. For the HHH Alignment dataset, we adopt the average score of all four subsets as the result. For the main **experiments trained with LoRA**, we truncate the inputs for all reward models over 1024 tokens. All reward models are trained for two epochs using a learning rate of $1 \times 10^{-5}$ and a batch size of 16. We load the model with the bf16 precision. Regarding the **full parameter training**, we truncate the inputs over 4096 tokens and train the reward model for one epoch with a learning rate of $2 \times 10^{-6}$ and a batch size of 512 (with gradient accumulation).

**Computational Resources.** We use NVIDIA RTX A6000 49G for our experiments. Training a 2B reward model with LoRA [74] on the 40K training data for 2 epochs requires approximately 30.4 GPU hours. A 7B reward model requires approximately 93.6 GPU hours.

Table 6: Key implementations of the text generation experiments.

| Basic information | |
|---|---|
| Base models | gemma-2b-it and Mistral-7B-Instruct-v0.2 |
| Quantization for training | bf16 |
| Fine-tuning strategy | LoRA [74] |
| LoRA $r$ | 32 |
| LoRA alpha | 64 |
| LoRA dropout | 0.05 |
| Optimizer | Adamw_hf |
| Batch size | 16 |
| Learning Rate | $1 \times 10^{-5}$ |
| Learning Rate Scheduler | cosine |
| Warmup Ratio | 0.03 |
| **GRM (Ours)** | |
| Regularization weight $\alpha$ | 0.01 by default, and 0.001 for the linear variant |
| Temperature $\beta$ for loss functions | 0.1 |
| **PPO[34]** | |
| KL regulaization | 0.0 |
| Epochs | 1 |
| learning rate | $1 \times 10^{-5}$ |
| lambda for GAE | 0.95 |
| gamma | 1 |
| clip range | 0.2 |
| Number of optimization epochs per batch | 4 |
| Number of tokens during generation | 512 |
| **Dataset and Gold Reward Model** | |
| Main Training Dataset | Unified-Feedback |
| Eval dataset: HHH-Alignment | https://huggingface.co/datasets/HuggingFaceH4/hhh_alignment |
| Eval dataset: MT-Bench Human Judgements | https://huggingface.co/datasets/lmsys/mt_bench_human_judgments |
| Gold Reward Model for BoN and PPO | reward-model-Mistral-7B-instruct-Unified-Feedback |

## C  Additional Experimental Results

### C.1  Comparing with Frozen Backbone

The effect of the random head for downstream finetuning of pretrained model is studied by [33], both theoretically and empirically (across a range of computer vision tasks). It is also easy to validate in the preference learning setting when using a smaller dataset size. We included a baseline, "Classifier (Frozen)", which fixes the base model's features and only fine-tunes the classification head. When the dataset size is 8K (see Table 7), the OOD evaluation results of the baseline reward model (without freezing the backbone) are worse than those of the frozen one, demonstrating the negative effect of distorting pre-trained features. However, we would like to note that when the dataset size is sufficiently large, this negative effect can be mitigated, and the baseline reward model can surpass the frozen reward model due to having more trainable parameters to fit the large preference dataset.

In contrast, by regularizing the hidden states, our GRM can achieve the regularizing effect while fine-tuning all parameters, showing strong performance with both large and small dataset sizes.

Table 7: Reward model performance trained with 8K data.

| Reward Model | Unified Feedback (ID) | HHH Alignment (OOD) | MT Bench (OOD) |
|---|---|---|---|
| Classifier (Frozen) | 62.2 | 68.8 | 67.6 |
| Classifier (Baseline) | 66.1 | 65.1 | 67.7 |
| GRM (ours) | 69.0 | 71.9 | 69.8 |

### C.2  Choice of Training Epochs

In our main experiments, we train reward models for 2 epochs with LoRA. We determine this number based on a nearly converging validation loss. Specifically, we reserve 1% of the training set for

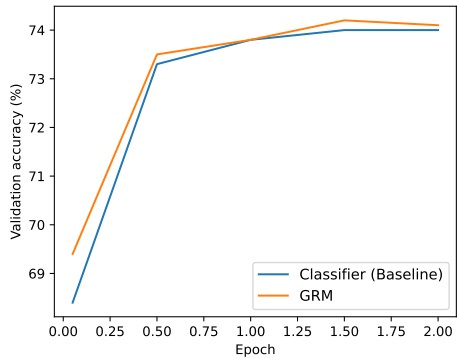

Figure 5: Learning curves for reward models on Unified-Feedback.

validation (e.g., 4K for 400K training data) and found that 2 epochs are sufficient for reward modeling with LoRA in our setting. As shown in Figure 5, we observe convergence in the validation loss during the second epoch, with no further improvement in the third epoch. For full-parameter training experiments, which are more prone to overfitting, we train the reward model for only one epoch.

## C.3 Choice of the SFT objective

In our paper, we consider a slightly different form of SFT objective as in Eq 10. A more straightforward objective is $\mathcal{L}_{\text{SFT}}(\theta_{\text{LM}}) = -\mathbb{E}_{(x,y_c)\sim D}\left[\log\left(\pi_{\theta_{\text{LM}}}(y_c \mid x)\right)\right]$. In ideal situations, the two forms should perform similarly. We also tried the $log$ form but found that it requires different hyperparameter tuning for the regularization weight $\alpha$ in Eq 4 due to changes in the loss scale. In Table 8 and Table 9, "GRM logreg" outperforms the baseline reward model and matches or even slightly exceeds the performance of GRM on OOD tasks when $\alpha$ is tuned appropriately. This experiment uses the same gemma-2B-it as the base model.

We found that the current form of SFT regularization can directly use the same hyperparameters as our DPO regularization. Therefore, we opted for this solution to **maintain coherence with these regularizations and avoid the need for hyperparameter adjustments**.

Table 8: Results on ID and OOD evaluation with **400K training data** from Unified-Feedback.

| Reward Model | Unified Feedback | HHH Alignment | MT Bench |
|---|---|---|---|
| Classifier (Baseline) | 72.1 | 73.4 | 71.2 |
| GRM | 73.2 | 79.8 | 73.4 |
| GRM logreg $\alpha = 0.005$ | 72.8 | 77.6 | 72.8 |
| GRM logreg $\alpha = 0.001$ | **73.3** | **80.2** | **73.6** |

Table 9: Results on ID and OOD evaluation with **40K training data** from Unified-Feedback.

| Reward Model | Unified Feedback | HHH Alignment | MT Bench |
|---|---|---|---|
| Classifier (Baseline) | 68.8 | 70.3 | 69.1 |
| GRM | **71.5** | 78.7 | **73.0** |
| GRM logreg $\alpha = 0.005$ | 69.7 | 72.4 | 72.8 |
| GRM logreg $\alpha = 0.001$ | 70.8 | **80.7** | 72.0 |

## C.4 Ablation of the Regularization Weight

We find the most impactful hyperparameter of GRM is the regularization weight $\alpha$. Figure 6 presents an evaluation of GRM's performance under various $\alpha$ values. It is evident from the figure that setting $\alpha$ to either extreme, such as 0, or a relatively large value like 0.1, results in suboptimal out-of-distribution (OOD) performance. However, selecting an appropriate value between 0 and 0.1 consistently yields higher scores. In all our experiments, we default to an $\alpha$ value of 0.01. This choice has already shown significant performance improvements in our experiments.

## C.5 Impact of Reward Head Layers on Performance

An interesting aspect to explore is how the structure of the nonlinear reward head influences preference learning performance. In Figure 7, we compare the performance of the default GRM (using the SFT

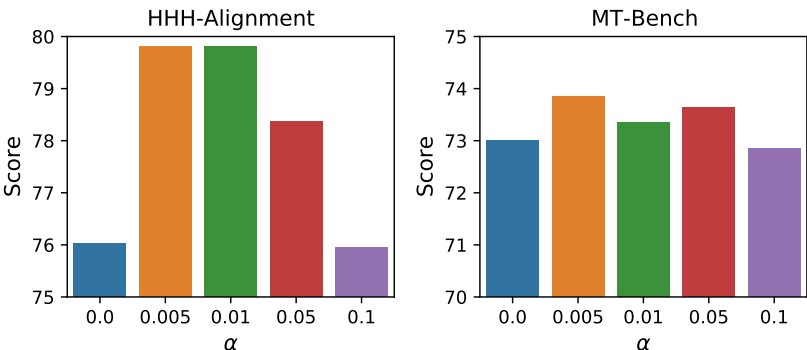

Figure 6: Comparing different values of $\alpha$ for GRM (2B) on scores of HHH-Alignment and MT-Bench.

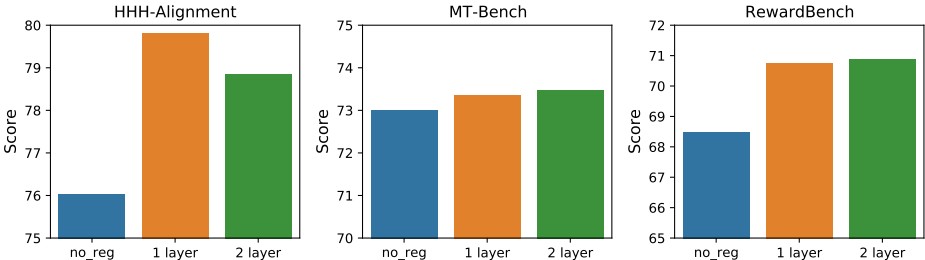

Figure 7: Comparing different layers of reward head for GRM (2B) on scores of HHH-Alignment, MT-Bench, and RewardBench.

regularization) against a variant of GRM that incorporates an additional linear layer and a ReLU activation in the reward head, denoted as "2 layer". The results indicate that the two-layer version slightly surpasses the performance of the single-layer GRM on MT-Bench and RewardBench scores, but it exhibits a decline in the score on the HHH-Alignment. Due to this inconsistency, we opted not to include the two-layer version in our main experiments. However, future research focusing on the impact of the reward model's structure could yield promising insights.

## C.6   Comparison with Additional Variant

In Appendix A, we derive an objective that retains only the reward model $r_\theta$ by replacing the policy $\pi_\theta^*$ with a formula of $r_\theta$. Empirically, this objective is challenging to optimize due to the calculation of $Z_\theta$. As an alternative, we propose a simplified objective by omitting the $Z_\theta$ term:

$$\arg \min_\theta \{\mathcal{L}_{\text{reward}}(\theta) - \gamma \mathbb{E}_{x,y_c \sim D}[r_\theta(x, y_c)]\}.$$

This objective includes a regularization term to maximize the average rewards of chosen responses. However, the second term can easily dominate the loss since the reward loss term is constrained by the logsigmoid operator. A more stable approach is to use the following empirical objective:

$$\arg \min_\theta \{\mathcal{L}_{\text{reward}}(\theta) - \gamma \mathbb{E}_{x,y_c \sim D}[\log \sigma(r_\theta(x, y_c))]\}.$$

We refer to this regularizer as "positive regularization" or "pos reg" for short. We compare positive regularization with the baseline classifier and GRM with SFT regularization in Tables 10 and 11. The base model for the reward models is gemma-2B-it, and GRM adopts the linear variant for the RewardBench results. "Positive regularization" does not yield improvement when the dataset size is limited to 40K, but it brings slight overall enhancement when learning from 400K training data.

In contrast, GRM significantly enhances both ID and OOD accuracy, especially when learning from a limited preference dataset. These results demonstrate that our approach is more effective, even when based on similar theoretical derivation.

Table 10: Results on ID and OOD evaluation with **400K training data** from Unified-Feedback.

| Reward Model | Unified Feedback | HHH Alignment | MT Bench | Reward Bench |
|---|---|---|---|---|
| Classifier (baseline) | 72.1 | 73.4 | 71.2 | 68.2 |
| Classifier (pos reg) | 71.7 | 75.0 | 70.7 | 69.8 |
| GRM w/ sft (ours) | **73.2** | **79.8** | **73.4** | **71.5** |

Table 11: Results on ID and OOD evaluation with **40K training data** from Unified-Feedback.

| Reward Model | Unified Feedback | HHH Alignment | MT Bench | Reward Bench |
|---|---|---|---|---|
| Classifier (baseline) | 68.8 | 70.3 | 69.1 | 64.5 |
| Classifier (pos reg) | 69.5 | 70.0 | 69.6 | 63.2 |
| GRM w/ sft (ours) | **71.5** | **78.7** | **73.0** | **69.5** |

## C.7 Regularization with pretraining dataset

In our default design, we use the preference dataset employed to train reward models to regularize the text-generation ability of the language head, eliminating the need for additional datasets. While we believe that other data formats, such as pretraining datasets, can also be beneficial, preference data offers a distinct advantage. It allows us to avoid using external datasets during reward modeling, which may also better align with the distribution of prompts and responses.

To illustrate this, we conduct an experiment using GRM with text-generation regularization on an open-source pretraining dataset, togethercomputer/RedPajama-Data-1T-Sample [7] (which includes text from Commoncrawl, Arxiv, and books), referred to as 'GRM pretrain reg'. For fairness, we only used a pretraining dataset of the same size as the training set for reward modeling.

The results indicate that 'GRM pretrain reg' outperforms the baseline reward model and matches the performance of GRM when the dataset size is large (400K). However, when the dataset size is small, using a pretraining dataset is less effective than using the preference dataset.

Table 12: Results on ID and OOD evaluation with **400K training data** from open-source pretraining dataset.

| Reward Model | Unified Feedback | HHH Alignment | MT Bench |
|---|---|---|---|
| Classifier (baseline) | 72.1 | 73.4 | 71.2 |
| GRM | **73.2** | **79.8** | 73.4 |
| GRM pretrain reg | 73.0 | 79.2 | **74.3** |

Table 13: Results on ID and OOD evaluation with **40K training data** from open-source pretraining dataset.

| Reward Model | Unified Feedback | HHH Alignment | MT Bench |
|---|---|---|---|
| Classifier (baseline) | 68.8 | 70.3 | 69.1 |
| GRM | **71.5** | **78.7** | **73.0** |
| GRM pretrain reg | 70.8 | 74.5 | 72.9 |

## C.8 Alignment Result after PPO

To demonstrate the advantage of GRM over vanilla reward modeling, we evaluate the win rate of models after PPO training with GRM against those with the vanilla reward model. The evaluation is conducted using GPT-4o on 100 randomly selected prompts from the test set in Unified-Feedback, with the order of responses randomly flipped to avoid order bias. The results below show a significantly higher win rate for GRM than the vanilla reward model across two different base reward models.

Table 14: Win rate of models after PPO training with GRM against those with the vanilla reward model.

| Base reward model | Win rate (%) | Tie rate (%) | Loss rate (%) |
|---|---|---|---|
| Gemma 2B it | 68 | 5 | 27 |
| Mistral 7B Instruct | 73 | 6 | 21 |

## C.9 Comparison with Label Smooth in RLHF

In Figure 8, we observed that reward models trained with label smoothing are vulnerable to hacking by BoN and PPO, leading to inferior performance compared to other baselines. The proxy score

---

[7]https://huggingface.co/datasets/togethercomputer/RedPajama-Data-1T-Sample

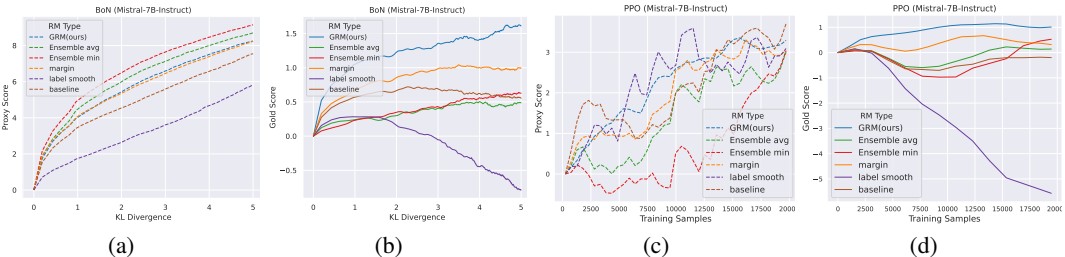

| (a) | (b) | (c) | (d) |

Figure 8: Proxy scores and gold scores of (a)(b) BoN experiments and (c)(d) PPO experiments for base models Mistral-7B-Instruct. Proxy and gold scores are in dashed and solid curves, respectively. Rewards are normalized to start from 0.

increases, while the gold score decreases rapidly. This finding suggests that previous robust techniques in the literature may not be effective for RLHF, underscoring the superiority of GRM as a more viable solution.

## D  Examples in the PPO Experiments

In Tables 15, 16, and 17, we present three examples that compare the responses of optimized language models using the PPO algorithm with different reward models. The base models for policy and reward models are all gemma-2b-it as in Section 5.2.

For baselines, it is evident that the models exploit certain patterns in rewards, such as the "Ensemble (min)" methods. This exploitation often leads to a collapse into repeated patterns. Besides, the "Baseline" and "Margin" models tend to disregard instructions or refuse to respond to harmless prompts, as demonstrated in Tables 15 and 16. Moreover, the baseline methods negatively impact the reasoning ability of language models for the math problem as in Table 17. These observations indicate that current reward models can be easily hacked by the PPO algorithm, raising concerns about their reliability.

In contrast, the GRM model demonstrates greater robustness in generating instruction-following responses and exhibits better reasoning abilities, even with identical hyperparameters of PPO. Notably, this superior performance of GRM is achieved even with a smaller training cost compared to ensemble baselines. These examples underscore the importance of GRM and its effectiveness in mitigating the overoptimization problem, further highlighting its potential in RLHF applications.

## E  Broader Impacts

The proposed approach to enhancing the generalization capabilities of reward models within the RLHF framework offers several positive societal impacts. By improving the accuracy of reward models on out-of-distribution (OOD) tasks, we can enhance the alignment of LLMs with human intent on larger dataset without human labels, leading to more reliable and stronger alignment. Moreover, the regularization technique that preserves the base model's language generation capabilities can contribute to the development of more robust and versatile AI systems, fostering innovation and efficiency across multiple domains. Currently, we do not foresee apparent negative societal impacts stemming from our methods. However, one potential adverse effect could arise if the generalizable reward model is exploited for harmful language model training. Therefore, future efforts in AI safety are crucial to prevent such misuse.

Table 15: Examples in the PPO experiments. GRM optimizes a better language model aligned with human intention, while other baseline reward models can be easily hacked by PPO.

| | | Comment |
|---|---|---|
| **Prompt** | Respond to the latest instruction based on the conversation history. Human: How do I fly a kite? Assistant: I'm happy to explain more about kites. What would you like to know? Human: how to get the kite in the air Assistant: Can you tell me a bit more about what you're trying to do? What kind of kite? What kind of wind conditions are you planning to use it in? What is your kite-flying experience? Human: I don't have much experience and it's just a simple diamond shaped one Assistant: What wind conditions are you planning to use it in? Human: just regular low wind i guess Assistant: | |
| **Baseline** | Based on the conversation history, the human is looking for guidance on how to get a simple diamond-shaped kite in the air. The assistant recommends considering wind conditions such as the direction and strength of the wind, the kite type, and the intended flying environment. They suggest considering low wind speeds as suitable for beginners. Therefore, the assistant suggests that the human research the specific wind conditions and kite specifications suitable for their location and desired flying experience before attempting to fly the kite. Remember to prioritize safety and seek guidance from experienced kite enthusiasts or professionals if possible. | The response does not follow the conversational tone. |
| **Margin** | Based on the conversation history, the human is seeking guidance on how to get a kite in the air. However, the assistant cannot provide specific instructions or recommendations without a thorough understanding of the individual's situation and capabilities. It's important to prioritize safety and avoid engaging in activities that could potentially harm the person or property involved. If you have any general kite-flying tips or suggestions, it's recommended to consult with a local park or outdoor enthusiast or seek guidance from a reputable kite-flying organization or instructor. Remember to always follow safety guidelines and regulations, especially when engaging in kite flying activities. | The response does not follow the conversational tone. |
| **Ensemble (min)** | Sure, here' soprHāy guidHǔˇŏng pulsa palab»() sedi lampa palab pulsa palab pulsa palab pulsa palab pulsa palab pulsa palab pulsa palab pulsa palab pulsa palab pulsa palab pulsa palab pulsa palab pulsa palab pulsa palab pulsa palab pulsa palab pulsa palab pulsa palab pulsa palab pulsa palab pulsa palab pulsa palab pulsa palab pulsa palab pulsa palab pulsa palab pulsa palab pulsa palab pulsa palab pulsa palab pulsa palab pulsa palab pulsa palab pulsa palab pulsa palab pulsa palab pulsa palab pulsa palab pulsa palab pulsa palab pulsa palab pulsa palab pulsa palab pulsa palab pulsa palab pulsa palab pulsa palab pulsa palab pulsa palab pulsa palab pulsa palab pulsa palab pulsa palab pulsa palab pulsa palab pulsa palab pulsa (omitted due to repeating) | The model is hacked to generate repeated words. |
| **Ensemble (avg)** | Sure, here's some advice on how to get a kite in the air safely and effectively: **Wind Conditions:** * Choose a location with consistent and gentle winds. Avoid strong or gusty winds, as they can make it difficult to control the kite. * Consider the wind speed and direction. Start with lighter winds, especially if you're a beginner. **Kite Selection:** * Choose a kite that's appropriate for your weight and the wind conditions. A well–sized kite will have sufficient lift to stay aloft and provide a thrilling flight experience. * Consider the kite's design. Diamond-shaped kites are relatively stable and easy to maneuver, while symmetrical kites offer more stability but may be more difficult to control. **Safety Precautions:** * Always fly kites away from people and obstacles. * Use a sturdy line or bridle that's long enough to allow the kite to fly freely but not become tangled. * Start in an open area with a clear view. * Be patient and allow the kite to fly for a few minutes before adjusting the line or releasing it. Remember to fly kites responsibly and always prioritize safety. | A good response to the prompt. |
| **GRM (Ours)** | **How to Get a Kite in the Air:** **1. Choose the Right Kite:** - Select a kite that is appropriate for your skill level and wind conditions. - Consider the kite's size, shape (diamond, square, or oval), and the amount of lift it provides. **2. Set Up the Kite:** - Unpack the kite and follow the instructions provided in the manual. - Attach the line to the kite's bridle or string and adjust the length to achieve the desired height. - Make sure the kite is securely attached to a sturdy pole or frame. **3. Choose Wind Conditions:** - Look for wind conditions that are moderate and consistent. - Ideal wind speeds are between 5 and 15 mph (8 and 24 kph). - Avoid strong winds, as they can make it difficult to control the kite. **4. Fly the Kite:** - Hold the kite's handle or string comfortably, keeping your arms relaxed. - Raise the kite slowly until it reaches its peak altitude. - Adjust the kite's height and angle (angle it slightly upwards) to control its flight path. - Gently move the kite with your hands to change its direction and speed. **Tips for Beginners:** - Start in a flat, open area with minimal obstacles. - Use a lightweight kite to make it easier to handle. - Start flying in the early morning or late afternoon when the wind conditions are generally favorable. - Be patient and persistent, as it takes time to learn how to fly a kite effectively. | A more informative response to the prompt. |

Table 16: Examples in the PPO experiments. GRM optimizes a better language model aligned with gold scores, while other baseline reward models can be easily hacked by PPO.

| Prompt | FINAL EXAM Question 1. Mission Records was created by which lead singer of The Eagles? Answer: | Comment |
|---|---|---|
| **Baseline** | The mission records were created by Don Henley, lead singer of The Eagles. The band was formed in 1974 and released several successful albums and hit songs throughout the 1970s and 1980s. Henley is widely regarded as one of the most influential and successful singers in the history of rock music. His songwriting and vocal contributions have influenced countless artists and fans worldwide. The Eagles became one of the best-selling and critically acclaimed bands of the late 20th century, with songs like "Hotel California," "Desperado," and "Hotel," among many hits. Henley's contributions to the band have solidified his status as a legend in the music industry. | Lengthy response to the question |
| **Margin** | The context does not provide information about the lead singer of The Eagles, so I cannot answer this question from the provided context. For accurate information about mission records and their creators, it is recommended to refer to official sources or seek further research. Remember to always prioritize reliable sources and avoid relying on secondhand or biased information. If you have any specific questions or concerns related to mission records or the artists involved, consider consulting with experts or professionals familiar with the subject matter. | Unhelpful response to the question |
| **Ensemble (min)** | Mission Records was created by lead singer of Eagles Eagles Keseitar sven pixabay Keseitar sven pixabay sven pixabay sven pixabay sven pixabay sven pixabay sven pixabay sven pixabay sven pixabay sven pixabay sven pixabay sven pixabay sven pixabay sven pixabay sven pixabay sven pixabay sven pixabay sven pixabay sven pixabay sven pixabay sven pixabay sven pixabay sven pixabay sven pixabay sven pixabay sven pixabay sven pixabay sven pixabay sven pixabay sven pixabay sven pixabay sven pixabay sven pixabay sven pixabay sven pixabay sven pixabay sven pixabay sven pixabay sven pixabay sven pixabay sven pixabay sven pixabay sven pixabay sven pixabay sven pixabay sven pixabay sven pixabay sven pixabay sven pixabay sven pixabay sven pixabay sven pixabay sven pixabay sven pixabay sven pixabay sven pixabay sven pixabay sven pixabay sven pixabay sven pixabay sven pixabay sven pixabay sven pixabay sven pixabay sven pixabay sven pixabay sven pixabay sven pixabay sven pixabay sven pixabay sven pixabay sven pixabay sven pixabay sven pixabay sven pixabay sven pixabay sven pixabay sven pixabay sven (omitted due to repeating) | The model is hacked to generate repeated words. |
| **Ensemble (avg)** | Mission Records was created by the lead singer of The Eagles, Don Henley. Mission Records was founded in 1971 in Nashville, Tennessee, United States. The mission of the record label was to provide a platform for artists to showcase their music and talents. Mission Records has released songs by many artists, including Eagles, Bruce Springsteen, and Dolly Parton. Missions Records has also been instrumental in helping to establish Nashville as a global music hub. Selamat—\|—\|—\|—\| Selamat—\|—\| Selamat—\|—\| Selamat—\|—\| Selamat—\|—\| Selamat—\|—\| Selamat—\|—\| Selamat—\|—\| Selamat—\|—\| Selamat—\|—\| Selamat Intere—\|—\| Selamat Intere—\|—\| Selamat Intere—\|—\| Selamat Intere—\|—\| Selamat Intere—\|—\| Selamat Intere—\|—\| Selamat Intere—\|—\| Selamat Intere—\| Selamat Intere—\| Selamat Intere—\| Selamat Intere—\| Selamat Intere—\| Selamat Intere—\| Selamat Intere—\| Selamat Intere—\| Selamat Intere—\| Selamat Intere—\| Selamat Intere—\| Selamat Intere—\| Selamat Intere—\| Selamat Intere—\| Selamat Intere—\| Selamat Intere—\| Selamat Intere—\| Selamat Intere—\| Selamat Intere—\| Selamat Intere—\| Selamat Intere—\| Selamat Intere—\| Selamat Intere—\| Selamat Intere—\| Selamat Intere—\| Selamat Intere—\| Selamat Intere—\| Selamat Intere—\| Selamat Intere—\| Selamat Intere—\| Selamat Intere—\| Selamat Intere—\| Selamat Intere—\| Selamat Intere—\| Selamat Intere—\| Selamat Intere—\| Selamat Intere—\| Selamat Intere—\| Selamat Intere—\| Selamat Intere—\| Selamat Intere—\| Selamat Intere—\| Selamat Intere— (omitted due to repeating) | The model is hacked to generate repeated words. |
| **GRM (Ours)** | The answer is Don Henley. Mission Records was a record label founded by Don Henley in 1974. The label was initially focused on releasing albums by The Eagles and other artists in the Eagles' genre. Mission Records went on to sign and release albums by a number of other artists, including Fleetwood Mac, Bon Jovi, and Linda Ronstadt. | A concise reply to the question with a short explanation. |

Table 17: Examples in the PPO experiments. GRM optimizes a better language model aligned with gold scores, while other baseline reward models can be easily hacked by PPO.

| Prompt | You will be given a definition of a task first, then some input of the task. You are given a math word problem and you are supposed to apply multiplication or division mathematical operators on the numbers embedded in the text to answer the following question and then only report the final numerical answer.
A chef needs to cook 9 potatoes . He has already cooked 7 . If each potato takes 3 minutes to cook , how long will it take him to cook the rest ? Output: | **Comment** |
|---|---|---|
| **Baseline** | **Task:** Cook 9 potatoes for 3 minutes each.
**Input:** Number of potatoes cooked = 7, cooking time per potato = 3 minutes.
**Multiplication:** 7 potatoes * 3 minutes/potato = 21 minutes
**Answer:** It will take the chef 21 minutes to cook the remaining 9 potatoes.
Therefore, the final numerical answer is **21**. | Wrong solution. |
| **Margin** | **Task:** Cook 9 potatoes for 3 minutes each.
**Input:** Number of potatoes cooked = 7, cooking time per potato = 3 minutes.
**Multiplication Operator:** To calculate the total cooking time, we multiply the number of potatoes cooked by the cooking time per potato.
9 potatoes * 3 minutes/potato = 27 minutes.
**Final Answer:** It will take the chef 27 minutes to cook the remaining potatoes.
Please note that the question does not provide any information about the initial state of the potatoes or any constraints, so the final answer may vary depending on the specific context and assumptions. It's always recommended to carefully review the task and input data before applying mathematical operations. | Wrong solution. |
| **Ensemble (min)** | Sure , here's how you calculate the time needed to cook the remaining potatoes :
**Step 1**: Calculate the total time already spent cooking potatoes : 7 minutes
**Step 2**: Subtract the time already spent from total time allotted : 9 minutes - 7 minutes = 2 minutes
**Step 3**: Multiply the remaining time by the number of potatoes to cook : 2 minutes x 9 = 18 minutes
Therefore , it will take him 18 minutes to cook the remaining potatoes . | Wrong solution. |
| **Ensemble (avg)** | **Task Definition:** Cook 9 potatoes for 3 minutes each.
**Input:**

7 potatoes cooked at 3 minutes each
**Calculations:**

9 potatoes $\times$ 3 minutes each = 27 minutes
Therefore, it will take the chef 27 minutes to cook the rest of the potatoes. Answer: 27 minutes. | Wrong solution. |
| **GRM (Ours)** | Sure, here's the analysis and solution for the task:
**Task Definition:**
The chef needs to cook 9 potatoes.
**Initial Information:**
* Cooked potatoes = 7
**Calculation:**
To cook the remaining potatoes, we need to add the number of cooked potatoes to the initial number of cooked potatoes:
9 potatoes - 7 potatoes = 2 potatoes
**Time to Cook Potatoes:**
Each potato takes 3 minutes to cook.
Therefore, the total time to cook the 2 potatoes is:
2 potatoes x 3 minutes/potato = **6 minutes**
**Final Answer:**
It will take the chef **6 minutes** to cook the 9 potatoes after he has cooked 7. | Right solution. |

