# OpenReview forum: "Regularizing Hidden States Enables Learning Generalizable Reward Model for LLMs"
_NeurIPS.cc/2024/Conference — NeurIPS 2024 poster_

### Official Review · Reviewer_VYhZ · 2024-06-16

**Soundness:** 3
**Presentation:** 3
**Contribution:** 4
**Rating:** 7
**Confidence:** 3

**Summary:**

This paper is concerned with the overoptimization issue in reward modeling: When optimizing a policy against a reward model, this leads to a distributional shift that can lead to an increase in the proxy score while the true score decreases. This paper addresses these issues by regularizing the hidden states in reward models utilizing an additional output head that (with some variations) predicts the text of the winning trajectories in the comparison data. The idea is that this prediction task (a) helps keep the intermediate, strong representations from pretraining, and (b) adapts them to text that is more similar to what's encountered in post-training, thus leading to stronger generalization to the data encountered when optimizing the policy.

**Strengths:**

My impression is that this is a strong paper dealing with the important overoptimization issue in RLHF. The motivation is strong and the experimental evaluation considers many baselines, settings, and datasets to be checked.

**Weaknesses:**

No strong weaknesses come to mind, though see the questions and comments below for other experiments that could be interesting to gain more confidence (though imho not necessary for this submission to be accepted).

**Questions:**

I'm using this section for questions, suggestions, and minor weaknesses.

a. "*In the RLHF stage, various policy optimization methods can be applied, with two frequently used methods being Best-of-n Sampling (BoN) and Proximal Policy Optimization (PPO).*" --- This paper seems to view RLHF as the stage *after* reward modeling, so just policy optimization. In contrast, I think it's more typical to view reward modeling as *part of* RLHF.

b. "*While straightforward, the DPO regularization requires a reference model during training.*" --- Don't you also need a reference policy during the PPO stage, given by the KL regularization with the SFT model? So it seems like this is no additional cost here?

c. Equation (7): Did you also consider replacing $\log \circ \sigma \circ \beta \circ \log$ by just $\log$? Do you expect this to perform better/worse?

d. Did you consider to interleave reward modeling with pretraining on a pretraining dataset to regularize the hidden states? I wonder if this performs better or worse than essentially doing pretraining on the preference dataset (as you do in your SFT regularization method).

e. Did you check whether in SFT regularization it matters whether you predict the text of the winning responses, instead of trying to predict the losing responses? I could imagine that the former performs better since it brings the reward model closer to understand a distribution that the PPO stage *will steer towards*, but I'm curious if that prediction holds up empirically.

f. For HHH-Alignment, MT-Bench, and RewardBench, could you give more details on what the benchmarks test for and what the scores mean?

g. I'd recommend putting the limitations and broader impact sections into the main paper.

**Limitations:**

Limitations are described in Appendix D.

---

> ### Author Rebuttal · Authors · 2024-08-06
>
> We thank the reviewer for the valuable comments, and we provide clarification to your concerns as follows. We appreciate it if you have any further questions or comments.
>
> **Q1:** This paper seems to view RLHF as the stage after reward modeling ... I think it's more typical to view reward modeling as part of RLHF.
>
> **A1:** Thanks for the question. We agree with the reviewer that reward modeling is one step of RLHF. We will revise Section 2 Background to reflect this more accurately.
>
>
> **Q2:** "The DPO regularization requires a reference model during training." --- Don't you also need a reference policy during the PPO stage ... So it seems like this is no additional cost here?
>
> **A2:** The cost includes both **memory usage and computation time**. Although DPO regularization does not increase memory usage for PPO, it still extends the computation time required for reward modeling. Furthermore, for other policy optimization methods such as BoN and reject sampling, the introduction of a reference model results in additional memory usage. Therefore, we claim that DPO regularization is more costly for reward modeling and prefer SFT and DPO w/o reference regularization according to the experimental results.
>
>
> **Q3:** Equation (7): Did you also consider replacing $log \sigma (\beta log)$  by just $log$? Do you expect this to perform better/worse?
>
> **A3:** In ideal situations, the two forms should perform similarly. We also tried the $log$ form but found that it requires different hyperparameter tuning for $\alpha$ due to changes in the loss scale. In the tables below, **"GRM logreg" outperforms the baseline reward model and matches or even slightly exceeds the performance of our GRM on OOD tasks when $\alpha$ is appropriate**.
>
> We found that the current form of SFT regularization can directly use the same hyperparameters as our DPO regularization. Therefore, we opted for this solution to maintain coherence with these regularizations and avoid the need for hyperparameter adjustments.
>
>
> | Reward Model (400K data)             | Unified Feedback (IID) | HHH Alignment (OOD) | MT Bench (OOD) |
> |----------------------------|------------------|---------------|----------|
> | Classifier (Baseline)      |  72.1 | 73.4 | 71.2 |
> | GRM |     73.2 | 79.8 | 73.4 |
> | GRM logreg $\alpha=0.005$ |    72.8  | 77.6 | 72.8 |
> | GRM logreg $\alpha=0.001$ |    **73.3**  | **80.2** | **73.6** |
>
>
>
> | Reward Model (40K data)             | Unified Feedback (IID) | HHH Alignment (OOD) | MT Bench (OOD) |
> |----------------------------|------------------|---------------|----------|
> | Classifier (Baseline)      |  68.8 | 70.3 |  69.1 |
> | GRM |     **71.5** | 78.7 | **73.0** |
> | GRM logreg $\alpha=0.005$ |  69.7   | 72.4 | 72.8|
> | GRM logreg $\alpha=0.001$ |   70.8   | **80.7**  | 72.0 |
>
>
>
> **Q4:** Did you consider to interleave reward modeling with pretraining on a pretraining dataset to regularize the hidden states?
>
> **A4:** Thanks for the insightful quesiton. One challenge of the described approach is that we don't have the pretraining dataset for many language models. We think that other data formats can also be useful as a regularization, but preference data is more advantageous because it allows us to avoid using external datasets during reward modeling by leveraging the preference dataset, which may better match the distribution of prompts and responses. Please refer to the **global response B** for the additional results.
>
>
> **Q5:** Did you check whether in SFT regularization it matters whether you predict the text of the winning responses, instead of trying to predict the losing responses?
>
> **A5:** In our SFT regularization, we only use the winning responses to regularize the hidden states. During the rebuttal, we also tested regularizing with the negative responses, referred to as "**GRM neg**." The results show **very close IID evaluation accuracy but relatively lower OOD accuracy** compared to our default setting, potentially due to the generally lower quality of rejected responses. The impact of this reward model on PPO training requires further examination. However, we agree with the reviewer's conjecture that the rejected responses may lead to unsatisfactory representations for the desired distribution.
>
>
> | Reward Model (400K data)             | Unified Feedback (IID) | HHH Alignment (OOD) | MT Bench (OOD) |
> |----------------------------|------------------|---------------|----------|
> | Classifier (Baseline)      |  72.1 | 73.4 | 71.2 |
> | GRM |     **73.2** | **79.8** | **73.4** |
> | GRM neg | **73.2**   |     78.1    |    73.0     |
>
>
>
> | Reward Model (40K data)             | Unified Feedback (IID) | HHH Alignment (OOD) | MT Bench (OOD) |
> |----------------------------|------------------|---------------|----------|
> | Classifier (Baseline)      |  68.8 | 70.3 |  69.1 |
> | GRM |     **71.5** | **78.7** | **73.0** |
> | GRM neg |  71.2   | 78.1 | 72.8|
>
>
>
> **Q6:** For HHH-Alignment, MT-Bench, and RewardBench, could you give more details on what the benchmarks test for and what the scores mean?
>
>
> **A6:** In Section 4, we briefly introduce three benchmarks. The HHH-Alignment dataset evaluates language models on helpfulness, honesty, and harmlessness. The MT-Bench dataset contains 3.3K human preferences for model responses generated by LLMs in response to MT-Bench questions. RewardBench is a new benchmark designed to evaluate the reward model's ability to select human-preferred responses for chat, reasoning, and safety tasks. In summary, these benchmarks are used to assess the reward model's alignment with human preferences and generalization ability across different prompt-response distributions. The scores represent the average accuracy across each benchmark, with higher accuracy indicating better performance.
>
>
> **Q7:** I'd recommend putting the limitations and broader impact sections into the main paper.
>
> **A7:** We agree with the reviewer and will move the two sections to the end of the main paper in the revised version.

---

> > ### Comment · Reviewer_VYhZ · 2024-08-08
> >
> > Thank you for the detailed answers to my questions!

---

> > > ### Author Response · Authors · 2024-08-09
> > >
> > > Thank you for your positive evaluation of our work. We greatly appreciate your insightful questions and suggestions, as they enhance the accuracy of our writing and the comprehensiveness of our evaluation.

---

### Official Review · Reviewer_SDk2 · 2024-06-24

**Soundness:** 3
**Presentation:** 3
**Contribution:** 3
**Rating:** 5
**Confidence:** 5

**Summary:**

This paper introduces a method that retains the base model's language model head while incorporating text-generation losses to preserve the hidden states' text generation capabilities.

**Strengths:**

* The paper is well-written, and the idea is straightforward.
* The code is easy to understand and implement.
* This method appears to be efficient and readily integrable with existing alignment approaches.

**Weaknesses:**

* Figure 3 (b) appears unusual, as the gold score decreases at the beginning of training. This could indicate suboptimal hyperparameter tuning or potential drawbacks in the pipeline.
* The results demonstrate limited advantages. It is recommended to validate the method's benefits with a larger dataset.

**Questions:**

* The training details mention that the reward models are trained for only 2 epochs. This choice may not be optimal, considering the risk of overfitting in the full fine-tuning setting. It would be helpful to discuss the reasoning behind this decision or the impact of learning rate and convergence speed.
* The simplicity of the regularization method is intriguing. Can it effectively mitigate reward hacking, potentially surpassing reward model ensemble techniques? It would be valuable to explore and discuss this aspect.

**Limitations:**

* The experiments may lack solidity due to the heavy reliance on the reward model in the PPO algorithm. If the reward model is not properly trained, the results can be significantly impacted, introducing randomness to the work.
* The inclusion of manually synthetic data in the dataset may not accurately reflect real-world data, limiting the generalizability of the findings.

---

> ### Author Rebuttal · Authors · 2024-08-06
>
> Thank you for the valuable comments, and we provide clarification to your concerns as follows.
>
> **Q1:** Figure 3 (b) appears unusual, as the gold score decreases at the beginning of training. This could indicate suboptimal hyperparameter tuning or potential drawbacks in the pipeline.
>
> **A1:** We found that this issue is influenced by **the learning rate of PPO**. By using a smaller learning rate (1e-5), we observed an initial improvement in the gold score during the first few steps, followed by subsequent drops (see **Figure 2 of the supplementary PDF**). Although the trend remains the same, the smaller learning rate delays the drop. In contrast, even with a larger learning rate, using our GRM allows PPO to learn stably, further emphasizing the importance of our proposed method for reward modeling and alleviating the burden of tuning hyperparameters for PPO.
>
> Moreover, we carefully followed the pipeline commonly adopted by the community [2][5] and used the open-source implementation of PPO from the Huggingface/trl package to ensure a correct and reliable pipeline.
>
>
>
> **Q2:** The results demonstrate limited advantages. It is recommended to validate the method's benefits with a larger dataset.
>
>
> **A2:** We would like to argue that the advantages of GRM are significant. As illustrated in Table 1 and 2 of our paper, increasing the training data from 40K to 400K enhances the OOD scores from 70.3/69.1 to 73.4/71.2. In contrast, GRM with just 40K training data achieves OOD scores of 78.7/73.0, showing a **much greater improvement than a 9x increase in the dataset size**. This is particularly important in real-world applications where there are limited samples for fine-tuning.
>
> Additionally, in **global response A**, we present further results on RewardBench, demonstrating the strong potential of scaling GRM to **larger models and datasets**. Our findings indicate that GRM even outperforms a 34B reward model and GPT-4 as a judge.
>
> Moreover, as shown in Table 1 and 2, GRM outperforms a strong ensemble baseline (n=3) while **using only about 1/3 of the computational cost**. In Section 5.2, GRM significantly outperforms other reward models for RLHF in terms of the gold score, alleviating the reward over-optimization problem of RLHF.
>
>
>
>
>
>
> **Q3:** The training details mention that the reward models are trained for only 2 epochs ... It would be helpful to discuss the reasoning behind this decision or the impact of learning rate and convergence speed.
>
>
> **A3:** In our experience, the training design depends on the dataset's quality, diversity, and size, as well as the base model and optimizer hyperparameters used for training. There is no one-size-fits-all hyperparameter setting. Prior work [2] trains reward models for 5 epochs and suggests **determining this number based on a nearly converging validation loss**.
>
> Following this insight, we reserve 1% of the training set for validation (e.g., 4K for 400K training data) and found that 2 epochs are sufficient for reward modeling with LoRA in our setting. As shown in the table below, we observed convergence in the validation loss during the second epoch, with no further improvement in the third epoch. For full-parameter training experiments added during rebuttal, which are more prone to overfitting, we train the reward model for only one epoch.
>
>
> | Validation accuracy (%) of reward model           | 0.05 epoch | 0.5 epoch | 1 epoch |  1.5 epoch   |   2 epoch      |
> |----------------------------|------------------|---------------|----------|------| -----|
> | Classifier (Baseline)      |  68.4 | 73.3 | 73.8 | 74.0 | 74.0 |
> | GRM |     69.4 | 73.5 | 73.8 |74.2 | 74.1|
>
>
>
> **Q4** The simplicity of the regularization method is intriguing. Can it effectively mitigate reward hacking, potentially surpassing reward model ensemble techniques?
>
>
> **A4:** Yes, in **Section 5.2**, we already explored the effectiveness of GRM for RLHF, particularly focusing on two commonly used policy optimization methods: BoN and PPO. As shown in Figure 3 of our main paper and 3 figures in the supplementary PDF, GRM consistently excels in both the PPO and noisy label experiments. Unlike the baselines (including the ensemble techniques), which show an increasing proxy score but a declining gold score, GRM demonstrates its effectiveness in mitigating reward hacking.
>
>
> **Q5:** The experiments may lack solidity due to the heavy reliance on the reward model in the PPO algorithm. If the reward model is not properly trained, the results can be significantly impacted, introducing randomness to the work.
>
>
> **A5:** We would like to emphasize that our main focus is on improving reward modeling (Section 5.1), subsequently enhancing RLHF (Section 5.2). **Our reward model is not limited to the PPO algorithm**; it can be applied to any policy optimization method based on explicit reward modeling, such as BoN or reject sampling. As the reviewer mentioned, an improperly trained reward model can be hacked by RL algorithms, as demonstrated in Section 5.2. In contrast, GRM shows better robustness against reward hacking, underscoring its importance for robust preference learning.
>
>
> **Q6:** The inclusion of manually synthetic data in the dataset may not accurately reflect real-world data, limiting the generalizability of the findings.
>
> **A6:** In Section 5.2, synthetic data is used in the RLHF experiment to allow the gold reward model to annotate preference labels without human labor. This approach is also adopted by prior works [2][3][5], and even research from companies like OpenAI and Google uses this synthetic method to reduce human labor. We have also mentioned this in the limitations section (Appendix D). In other experiments in **Section 5.1**, we use the UnifiedFeedback dataset, which includes **datasets based on human annotators**, such as Anthropic/hh-rlhf, lmsys/chatbot_arena_conversations, openai/summarize_from_feedback. Therefore, we believe our setting also reflects real-world data.

---

> > ### Comment · Reviewer_SDk2 · 2024-08-11
> >
> > Thanks for the authors' comprehensive response. I would like to keep the current score. Thanks!

---

> > > ### Author Response · Authors · 2024-08-11
> > >
> > > Thank you for your time, effort, and valuable feedback. We hope that our responses, including detailed explanations regarding the hyperparameter design and validation on a larger dataset, have addressed your concerns.
> > >
> > > If our responses have resolved your issues, we would greatly appreciate it if you could consider raising the score. If you have any additional concerns, please feel free to post them, and we would be happy to discuss them further before the discussion deadline on August 13th.

---

> > > > ### Comment · Reviewer_SDk2 · 2024-08-12
> > > >
> > > > Yes, after carefully reviewing your response, I have decided to raise the score to 5. While the paper's innovation appears limited, I want to emphasize the positive aspects, particularly the strong experimental results.

---

> > > > > ### Comment · Reviewer_VYhZ · 2024-08-12
> > > > >
> > > > > (I'm another reviewer chiming in):
> > > > >
> > > > > Could it be that you misclicked and accidentally raised your confidence instead of the score to 5?

---

> > > > > ### Author Response · Authors · 2024-08-12
> > > > > **Thanks for raising the score!**
> > > > >
> > > > > We would like to thank the reviewer for acknowledging the positive aspects of our work. We greatly appreciate your valuable comments and suggestions, which enhance the quality of our paper.

---

### Official Review · Reviewer_NAqK · 2024-07-07

**Soundness:** 3
**Presentation:** 3
**Contribution:** 3
**Rating:** 7
**Confidence:** 4

**Summary:**

This paper proposes generalizable reward model (GRM) which modifies the standard reward-learning objective by adding an auxiliary task with a separate language modeling head. The auxiliary loss is either DPO or SFT. Experiments and ablations are conducted using mistral and gemma models and data from unified-feedback and evaluating on HHH, MT-bench, and rewardbench showing GRM improves OOD performance. Finally, GRM reward models are used to train new policies.

**Strengths:**

1. The idea of adding an auxiliary loss is simple and elegant. The paper provides a good intuitive argument for why it may be useful and clearly presents the potential ways to implement the auxiliary loss (DPO and SFT).

2. The experiments and ablations are thorough and well-motivated. The paper considers a variety of baseline techniques to make reward models more robust like adding a margin term, label smoothing, or an ensemble. The paper also does a good job of comparing the low-data and high-data regimes.

3. The results seem to show a consistent, if sometimes modest, improvement from GRM over baseline methods. And the improvements are larger in the low-data regime.

**Weaknesses:**

1. While the paper is generally comprehensive, there could be additional exploration of just using language modeling as an auxiliary task. It is not clear if the benefit of GRM is coming from language modeling the data from the preference dataset or if it would benefit from language modeling on any data (or even data from an even broader distribution, controlling for more OOD data). A further exploration of this would be good.

**Questions:**

None

**Limitations:**

There could be more discussion of limitations

---

> ### Author Rebuttal · Authors · 2024-08-06
>
> Thank you for the valuable comments, and we provide clarification to your concerns as follows.
>
> **Q1:** While the paper is generally comprehensive, there could be additional exploration of just using language modeling as an auxiliary task. It is not clear if the benefit of GRM is coming from language modeling the data from the preference dataset or if it would benefit from language modeling on any data (or even data from an even broader distribution, controlling for more OOD data). A further exploration of this would be good.
>
>
> **A1:** Thanks for the insightful question. We believe that other data formats can also be useful, but preference data is more advantageous because it allows us to avoid using external datasets during reward modeling by leveraging the preference dataset, which may better match the distribution of prompts and responses. To illustrate this, we provide an experiment below using GRM with text-generation regularization on an open-source pretraining dataset, togethercomputer/RedPajama-Data-1T-Sample (including text from Commoncrawl, Arxiv, and books), referred to as 'GRM pretrain reg'. The results in **global response B** suggest that 'GRM pretrain reg' outperforms the baseline reward model and matches the performance of GRM when the dataset size is large (400K). However, when the dataset size is small, using a pretraining dataset is less effective than using the preference dataset.
>
>
> **Q2:** There could be more discussion of limitations
>
>
> **A2:** Thank you for the question. In addition to the limitations mentioned in Appendix D, another constraint is that we did not test reward models with parameter sizes exceeding 10B due to computational limitations. In the main paper, we tested models with sizes of 2B and 7B. During the rebuttal period, we also provided results for 8B reward models in **global response A**, which demonstrate promising potential for scaling GRM to larger reward models and datasets. Further efforts to extend our method to even larger reward models could be highly promising.

---

> > ### Comment · Reviewer_NAqK · 2024-08-09
> >
> > Thanks for your response and for running the additional experiments. Indeed it is interesting that using pre-training data for regularization can also be effective, while not quite as effective as the original method.
> >
> > I will leave my accept score and continue to think this is a strong paper.

---

> > > ### Author Response · Authors · 2024-08-09
> > >
> > > Thank you for your positive evaluation of our work. We greatly appreciate the insightful questions, particularly the one regarding the use of other text data for regularization, as they help make our work more comprehensive.

---

### Official Review · Reviewer_tSHJ · 2024-07-13

**Soundness:** 3
**Presentation:** 3
**Contribution:** 2
**Rating:** 4
**Confidence:** 4

**Summary:**

The paper addresses the limitations of current reward models used in the reinforcement learning from human feedback (RLHF) framework, specifically their generalization capabilities to unseen prompts and responses. This limitation often leads to reward over-optimization, where the excessive optimization of rewards results in a decline in actual performance. The study proposes an approach to enhance the reward model's generalization ability against distribution shifts by regularizing the hidden states.

**Strengths:**

The motivation for the study is sound, and the experiments validate the effectiveness of the proposed method.

**Weaknesses:**

1. The proposed method lacks innovation as it combines DPO and SFT loss into the RM training phase as a regularization term. The inclusion of SFT loss in RM training has already been explored in previous works, such as InstructGPT and Anthropic's RM training.

2. In the introduction, the authors mention that a randomly initialized head can distort pre-trained features, negatively impacting out-of-distribution (OOD) performance. Inspired by this finding, they propose to regularize feature distortion during fine-tuning for preference learning. However, there is no experimental evidence provided to support that this motivation holds true for RM. The improvements from adding regularization alone do not sufficiently prove that this motivation is solid.

3. The results for label smoothing are missing in Figures 2 and 3, which should be addressed.

4. The experimental section should include the alignment results after RL.

**Questions:**

Please refer to the Weaknesses section for questions regarding the paper.

**Limitations:**

Yes

---

> ### Author Rebuttal · Authors · 2024-08-06
>
> Thank you for the valuable comments, and we provide clarification to your concerns as follows.
>
>
> **Q1**: The proposed method lacks innovation ... The inclusion of SFT loss in RM training has already been explored in previous works, such as InstructGPT and Anthropic's RM training.
>
> **A1:** We would like to clarify that **no previous work has incorporated text-generation loss as a regularization term for reward modeling**. InstructGPT and Anthropic's RM training optimize the vanilla reward loss (Eq 1 in [1]) based on the Bradley-Terry model, and their SFT regularization is typically applied to **language model training rather than reward modeling**.
>
> Our method is novel in that it integrates text-generation loss directly into the reward modeling process, a fundamentally different task from text generation. The enhancement effect of text-generation loss on reward modeling has not been explored in prior work. Additionally, our reward model structure is distinct to enforce this regularization. We maintain both a language head and a reward head within the reward model and apply text-generation loss to the language head during reward training.
>
>
> In our experiments, compared to previous methods that use ensemble techniques [2][3] to enhance reward model capabilities, our method significantly reduces computational costs while achieving superior performance. This demonstrates promising results for more reliable and cost-effective reward modeling.
>
>
>
> **Q2**: No experimental evidence supporting the claim that a randomly initialized head can distort pre-trained features in RM, consequently leading to a negative impact on OOD performance.
>
>
> **A2:** The phenomenon is well-documented by [4], both theoretically and empirically (across a range of computer vision tasks). It is also easy to validate in the preference learning setting when using a smaller dataset size. We included a baseline, "Classifier (Frozen)", which fixes the base model’s features and only fine-tunes the classification head. When the dataset size is 8K (see the table below), the OOD evaluation results of the baseline reward model (without freezing the backbone) are worse than those of the frozen one, demonstrating the negative effect of distorting pre-trained features. However, we would like to note that when the dataset size is sufficiently large, this negative effect can be mitigated, and the baseline reward model can surpass the frozen reward model due to having more trainable parameters to fit the large dataset.
>
>
> In contrast, by regularizing the hidden states, our GRM can achieve the regularizing effect while fine-tuning all parameters, showing strong performance with both large and small dataset sizes.
>
>
> | Reward Model   (8K data)           | Unified Feedback (IID) | HHH Alignment (OOD) | MT Bench (OOD) |
> |----------------------------|------------------|---------------|----------|
> | Classifier (Frozen)        |    62.2        |   68.8       | 67.6   |
> | Classifier (Baseline)      |  66.1   |  65.1    |  67.7 |
> | GRM (ours)|   **69.0**     | **71.9**           |       **69.8**    |
>
>
>
> **Q3**: The results for label smoothing are missing in Figures 2 and 3, which should be addressed.
>
>
> **A3:** We found that reward models with label smoothing are easily hacked by BoN and PPO, resulting in worse performance compared to other baselines. Therefore, we omitted them from the policy optimization experiments. To address the reviewer's concern, we have included the label smoothing results for the BoN and PPO experiments using the 7B base model in **Figure 1 of the supplementary PDF** provided during the rebuttal.
>
>
> **Q4**: The experimental section should include the alignment results after RL.
>
> **A4:** In Section 5.2, we evaluated different reward models for both RL training and Best-of-N sampling, using the gold score as the measurement for alignment, as adopted by prior studies [2][5] for reward modeling. Additionally, to demonstrate the advantage of GRM over vanilla reward modeling, we evaluated **the win rate of models after PPO training with GRM against those with the vanilla reward model**. The evaluation was conducted using GPT-4o on 100 randomly selected prompts from the test set in UnifiedFeedback, with the order of responses randomly flipped to avoid order bias. The results below show a significantly higher win rate for GRM than the vanilla reward model across two different base reward models.
>
>
> | Base reward model       | Win rate  | Tie rate | Loss rate |
> |----------------------------|------------------|---------------|----------|
> |   Gemma 2B it|     0.68    |    0.05   |  0.27  |
> | Mistral 7B Instruct         | 0.73   |  0.06      | 0.21 |

---

> ### Author Response · Authors · 2024-08-12
> **Looking Forward to Your Valuable Feedback**
>
> Dear Reviewer tSHJ,
>
> We deeply appreciate your time, effort, and valuable feedback. We hope that our responses, including detailed explanations about the differences between the prior reward modeling paradigm and our approach, as well as additional experimental results on feature distortion and alignment after RL, have addressed your concerns.
>
> If our responses have resolved your issues, we would greatly appreciate it if you could consider raising the score. If you have any additional concerns, please feel free to post them, and we would be happy to discuss them further before the discussion deadline on Aug 13th.
>
> Thank you once again for your thoughtful review.
>
> Best,
> The Authors

---

> ### Author Response · Authors · 2024-08-14
> **Looking Forward to Your Feedback**
>
> Dear Reviewer tSHJ,
>
> We deeply appreciate your time, effort, and valuable feedback. **Since the discussion deadline is less than 10 hours away and we have not yet received an acknowledgment from you, we kindly request you to review our replies**. We hope that our responses, including detailed explanations about the differences between the prior reward modeling paradigm and our approach, as well as additional experimental results on feature distortion and alignment after RL, have addressed your concerns.
>
>  If our responses have resolved your issues, we would greatly appreciate it if you could consider raising the score. If you have any additional concerns, please feel free to post them, and we would be happy to discuss them further before the discussion deadline
>
> Thank you once again for your constructive review.
>
> Best,
>
> The Authors

---

### Author Rebuttal · Authors · 2024-08-06

We thank all reviewers for their constructive comments and are particularly grateful for their recognition of our work: 'sound and strong motivation' (Reviewers tSHJ, VYhZ), 'elegant idea' (Reviewer NAqK), 'thorough experiments' (Reviewers NAqK, VYhZ), and 'well-written' (Reviewer SDk2). We hope our responses have addressed all of the reviewers' concerns. If you have any additional questions, please feel free to post them, and we will be happy to discuss further.

Below, we provide new experimental results and global references.

### A. **8B full parameter finetuning on larger dataset**

To further demonstrate the effectiveness of GRM, we present additional results on RewardBench, a new benchmark for human preference. We trained GRM using the llama3-8b-instruct model, performing full parameter fine-tuning for 1 epoch on one of the largest open-source preference datasets, hendrydong/preference_700K. Our results indicate the **strong potential of scaling GRM to larger models and datasets**, even outperforming a 34B reward model and GPT-4 as a judge.


|       Reward model           | Average       |  Chat     |     Chat Hard      |     Safety      |     Reasoning     |
|:-------------------------:|:-------------:|:---------:|:---------:|:--------:|:-----------:|
|  GRM (Ours, 8B) | 87.0     |   98.6  |  67.8 |   89.4 |92.3     |
|    openai/gpt-4-0125-preview                             |    85.9     |   95.3      |  74.3  |  87.2 |    86.9    |
|openai/gpt-4-turbo-2024-04-09 | 85.1 |  95.3   | 75.4   |87.1 | 82.7|
|   Classifier   (baseline, 8B)                          |   	84.7     |   99.4     |   65.1   | 87.8  |    86.4  |
|Nexusflow/Starling-RM-34B | 82.7 | 96.9 |57.2|88.2 |88.5|


### B. **Regularization with pretraining dataset**

In our default design, we use the preference dataset employed to train reward models to regularize the text-generation ability of the language head, eliminating the need for additional datasets. While we believe that other data formats, such as pretraining datasets, can also be beneficial, preference data offers a distinct advantage. It allows us to avoid using external datasets during reward modeling, which may also better align with the distribution of prompts and responses.

To illustrate this, we conducted an experiment using GRM with text-generation regularization on an open-source pretraining dataset, togethercomputer/RedPajama-Data-1T-Sample (which includes text from Commoncrawl, Arxiv, and books), referred to as **'GRM pretrain reg'**. For fairness, we only used a pretraining dataset of the same size as the training set for reward modeling.

The results indicate that 'GRM pretrain reg' outperforms the baseline reward model and matches the performance of GRM when the dataset size is large (400K). However, when the dataset size is small, using a pretraining dataset is less effective than using the preference dataset.

| Reward Model (400K data)             | Unified Feedback (IID) | HHH Alignment (OOD) | MT Bench (OOD) |
|----------------------------|------------------|---------------|----------|
| Classifier (Baseline)      |  72.1 | 73.4 | 71.2 |
| GRM |     **73.2** | **79.8** | 73.4 |
| GRM pretrain reg  |   73.0  | 79.2 | **74.3** |




| Reward Model (40K data)             | Unified Feedback (IID) | HHH Alignment (OOD) | MT Bench (OOD) |
|----------------------------|------------------|---------------|----------|
| Classifier (Baseline)      |  68.8 | 70.3 |  69.1 |
| GRM |     **71.5** | **78.7** | **73.0** |
| GRM pretrain reg |  70.8   | 74.5 | 72.9|


### C. **Suplementary PDF**

In our supplementary PDF for the rebuttal, we include three figures:

- **Figure 1**: We present the results of label smoothing to address reviewer tSHJ's concern.
- **Figure 2**: We provide the results of PPO with a learning rate of $1 \times 10^{-5}$ to address reviewer SDk2's concern about the hyperparameter of PPO.
- **Figure 3**: We include the results of PPO in a noisy label setting. These results consistently demonstrate the superiority of GRM compared to other reward model baselines.


### D. **References**

[1] Training language models to follow instructions with human feedback. NeurIPS, 2022.

[2] Reward Model Ensembles Help Mitigate Overoptimization. ICLR 2024.

[3] Warm: On the benefits of weight averaged reward models. ICML, 2024.

[4] Fine-Tuning can Distort Pretrained Features and Underperform Out-of-Distribution. ICLR 2022.

[5] Scaling laws for reward model overoptimization. ICML, 2023.

---

### Decision · Program_Chairs · 2024-09-25

**Decision:**

Accept (poster)

**Comment:**

The paper studies the issue of reward generalization. It re-introduces the language model loss to reward model training as a regularization term to improve reward generalization. Experiment results show that this method could improve the reward model performance on multiple benchmarks. Reviews generally agree that the motivation for the study is sound, and the experiments validate the effectiveness of the proposed method.